# Nucleotide imbalance decouples cell growth from cell proliferation

Frances F. Diehl[1,2], Teemu P. Miettinen[1,3], Ryan Elbashir[1,2], Christopher S. Nabel[1,4], Alicia M. Darnell[1,2], Brian T. Do[1,2,5], Scott R. Manalis[1,6], Caroline A. Lewis[7] and Matthew G. Vander Heiden [1,2,8] ✉

Nucleotide metabolism supports RNA synthesis and DNA replication to enable cell growth and division. Nucleotide depletion can inhibit cell growth and proliferation, but how cells sense and respond to changes in the relative levels of individual nucleotides is unclear. Moreover, the nucleotide requirement for biomass production changes over the course of the cell cycle, and how cells coordinate differential nucleotide demands with cell cycle progression is not well understood. Here we find that excess levels of individual nucleotides can inhibit proliferation by disrupting the relative levels of nucleotide bases needed for DNA replication and impeding DNA replication. The resulting purine and pyrimidine imbalances are not sensed by canonical growth regulatory pathways like mTORC1, Akt and AMPK signalling cascades, causing excessive cell growth despite inhibited proliferation. Instead, cells rely on replication stress signalling to survive during, and recover from, nucleotide imbalance during S phase. We find that ATR-dependent replication stress signalling is activated during unperturbed S phases and promotes nucleotide availability to support DNA replication. Together, these data reveal that imbalanced nucleotide levels are not detected until S phase, rendering cells reliant on replication stress signalling to cope with this metabolic problem and disrupting the coordination of cell growth and division.

Most proliferating cells double each component of their mass over the course of the cell cycle; metabolic demands therefore shift to enable biosynthetic processes specific to different cell cycle phases[1,2]. Proliferating cells have a particularly high demand for nucleotides and must acquire sufficient levels of each nucleotide species both for RNA synthesis and to ensure efficient and accurate DNA replication during S phase. Nucleotides are required for ribosomal RNA and messenger RNA synthesis to enable biomass production. RNA production contributes to biomass both directly, as RNA accounts for the vast majority of nucleic acid in cells[3], and indirectly by enabling protein production. Nucleotide acquisition is therefore essential not only for cell cycle progression and division, but also for biomass synthesis to enable cell growth. This raises the question of how cells coordinate differential needs for nucleotides in supporting cell growth and enabling genome replication specifically during S phase.

Ribonucleotide reductase (RNR) mediates dNTP production, converting ribonucleoside diphosphates to deoxyribonucleoside diphosphates. RNR inhibition impairs DNA replication and induces replication stress signalling[4], indicating that a substrate-level limitation for dNTPs can impede DNA synthesis. In budding yeast, insufficient dNTPs at the onset of S phase can activate replication stress signalling in unperturbed cells[5], suggesting that endogenous dNTP levels are within a range that can become limiting. Further, RNR mutations that lead to depletion of specific dNTPs can slow S phase progression in budding yeast[6], underscoring the importance of maintaining appropriate levels of individual dNTPs for DNA replication.

Cells have evolved conserved signalling networks that match growth with metabolic capacity by coordinating responses to stress conditions and nutrient availability[1]. Upon nutrient limitation, growth control pathways generally arrest cell growth and down-regulate biosynthesis to preserve resources[1,7,8]. Growth signalling plays a role in nucleotide metabolism, both by coordinating RNA production and breakdown and by regulating de novo nucleotide synthesis. For example, the mTORC1 substrate p70 S6 kinase phosphorylates and stimulates a key enzyme in pyrimidine synthesis[9], and mTORC1 signalling promotes production of one-carbon substrates for purine synthesis[10]. Nucleotide availability can also be an important input for growth control pathways, and purine levels regulate mTORC1 activity[11,12]. In cells with defective autophagy, providing nucleotides alone allows survival in starvation conditions[13], highlighting the importance of nucleotide homeostasis for cellular fitness across environmental conditions.

Different nucleotide species have distinct roles in cell metabolism and vary over a wide range of intracellular concentrations[3,14]. Extracellular nucleobase and nucleoside availability varies on the basis of physiological context[1,15], but these species are often environmentally scarce. Accordingly, while many cells preferentially salvage available nucleobases and nucleosides, most cells must rely on de novo synthesis to fulfil at least part of their nucleotide demands. Both purine synthesis and pyrimidine synthesis involve multiple metabolic pathways that can be differentially affected by environmental perturbations[1,16–18]. Thus, environmental availability of nutrients, including nucleotide precursors, can affect relative levels of individual nucleotides in cells. It remains unclear whether cells sense the relative availability of specific nucleotide species and how cells maintain nucleotide homeostasis to meet shifting demands throughout the cell cycle.

In this Article, we show that imbalances among nucleotide species inhibit cell proliferation but are not sensed by canonical

[1]Koch Institute for Integrative Cancer Research, Massachusetts Institute of Technology, Cambridge, MA, USA. [2]Department of Biology, Massachusetts Institute of Technology, Cambridge, MA, USA. [3]Medical Research Council Laboratory for Molecular Cell Biology, University College London, London, UK. [4]Massachusetts General Hospital Cancer Center, Boston, MA, USA. [5]Harvard–MIT Health Sciences and Technology, Cambridge, MA, USA. [6]Departments of Biological Engineering and Mechanical Engineering, Massachusetts Institute of Technology, Cambridge, MA, USA. [7]Whitehead Institute for Biomedical Research, Cambridge, MA, USA. [8]Dana–Farber Cancer Institute, Boston, MA, USA. ✉e-mail: mvh@mit.edu

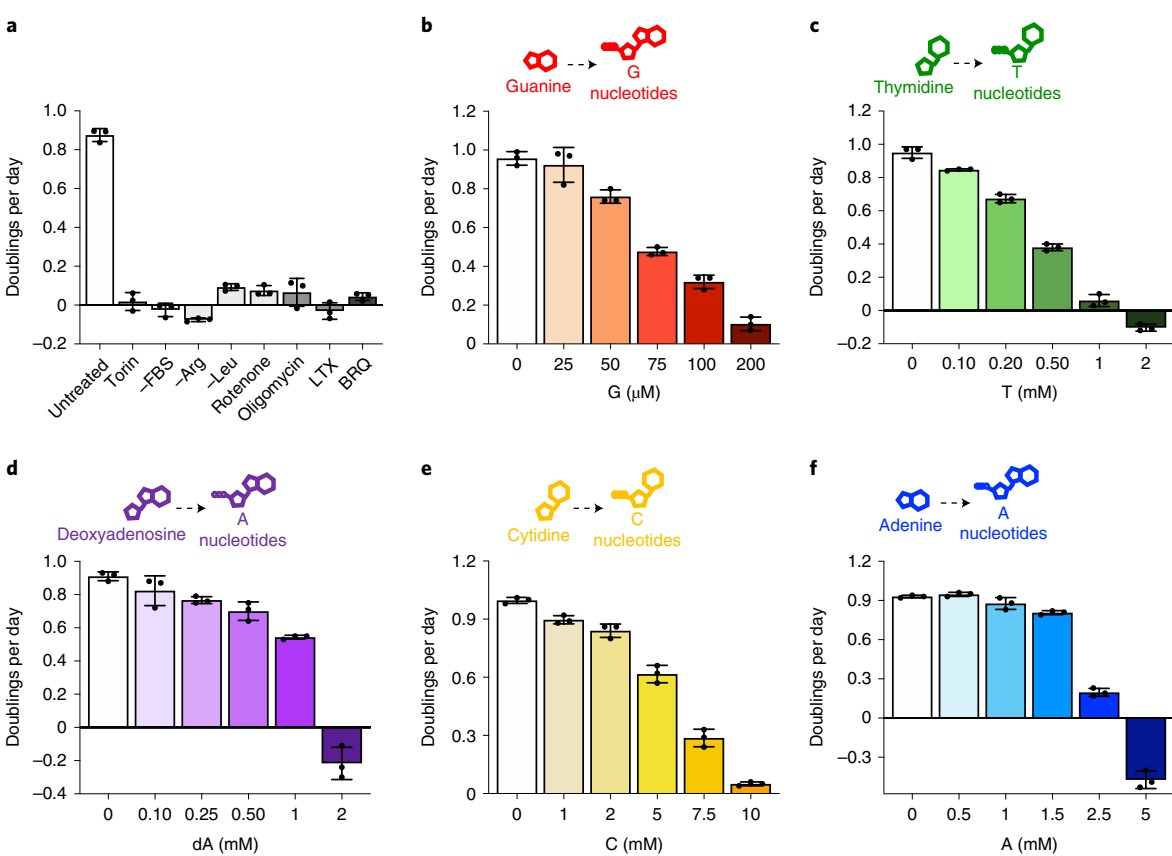

**Fig. 1 | Excess individual nucleotides can impair proliferation. a,** Proliferation rates of A549 cells cultured in standard conditions (untreated) or with 1 μM Torin1, without serum (−FBS), without arginine (−Arg), without leucine (−Leu), with 100 nM rotenone, with 5 nM oligomycin, with 1 μM lometrexol (LTX) or with 1 μM brequinar (BRQ). **b–f,** Proliferation rates of A549 cells treated with the indicated concentration of guanine (G), thymidine (T), deoxyadenosine (dA), cytidine (C) or adenine (A). Each of these nucleobases/nucleosides can be salvaged to produce intracellular nucleotides as shown. Data are presented as mean ± standard deviation (s.d.) of three biological replicates. Source numerical data are available in source data.

metabolic regulatory pathways. Rather, cells continue to grow and enter S phase despite nucleotide imbalance, leading to activation of DNA replication stress signalling as a protective response. Replication stress signalling also promotes nucleotide availability during unperturbed S phases, suggesting that replication stress sensing may play a role in sensing and maintaining nucleotide balance during normal proliferation.

## Results

**Nucleotide precursors can inhibit cell proliferation.** Diverse metabolic perturbations can inhibit cell proliferation, including disruptions to pro-growth signalling pathways, amino acid availability, mitochondrial respiration and nucleotide synthesis (Fig. 1a). Obtaining nucleotides can be particularly limiting for cell proliferation[19–21]. Indeed, pharmacological inhibition of purine production with lometrexol (LTX) or pyrimidine production with brequinar (BRQ) depletes total purine or pyrimidine levels and blocks proliferation, consistent with previous studies[11,22,23] (Fig. 1a and Extended Data Fig. 1a). Intriguingly, thymidine treatment has long been used to arrest and synchronize cells; however, the proximal mechanisms of this arrest, and whether this has broader implications for the regulation of nucleotide homeostasis, are less clear. Moreover, thymidine exists uniquely in the dNTP pool, and it is unclear whether perturbations to ribonucleotide (NTP) pools are equally detrimental. To investigate this, we supplemented cells with individual nucleobases and nucleosides, which can be salvaged to produce nucleotides. Nucleotide salvage preserves metabolic substrates that would otherwise be needed for de novo nucleotide synthesis. However, we found

that single nucleobase or nucleoside supplementation impaired proliferation in a dose-titratable manner (Fig. 1b–f).

Expression of nucleotide salvage and synthesis enzymes, as well as transporters, varies across cells and could affect sensitivity to individual nucleobase or nucleoside addition. Consistent with this, different cells had differential sensitivity to each species, although nucleotide precursor addition could inhibit proliferation of all cells tested, including non-transformed cells (Extended Data Fig. 1b,c). Interestingly, deoxycytidine (at concentrations up to 14 mM) was the only precursor tested that did not inhibit cell proliferation (Extended Data Fig. 1d). As most cells tested exhibited greatest sensitivity to guanylate nucleotide precursors (Extended Data Fig. 1e), we focused further mechanistic studies on understanding the effects of guanine supplementation. Importantly, a functional salvage pathway was needed for the corresponding nucleotide precursor to inhibit proliferation: cells deficient for APRT and HPRT, the enzymes that salvage adenine and guanine, were not sensitive to these precursors, and thymidine kinase-deficient 143B cells were unaffected by thymidine addition (Extended Data Fig. 1b,f).

**Nucleotide imbalance impairs cell proliferation.** We reasoned that salvage of single nucleobases/nucleosides might perturb relative levels of intracellular nucleotide species and measured nucleotide levels in cells with or without guanine addition. Guanine supplementation increased intracellular levels of guanylate (G) nucleotides (GTP/GDP/GMP) and unexpectedly decreased intracellular levels of adenylate nucleotides (A) (ATP/ADP/AMP) (Fig. 2a and Extended Data Fig. 2a). These data suggest that guanine

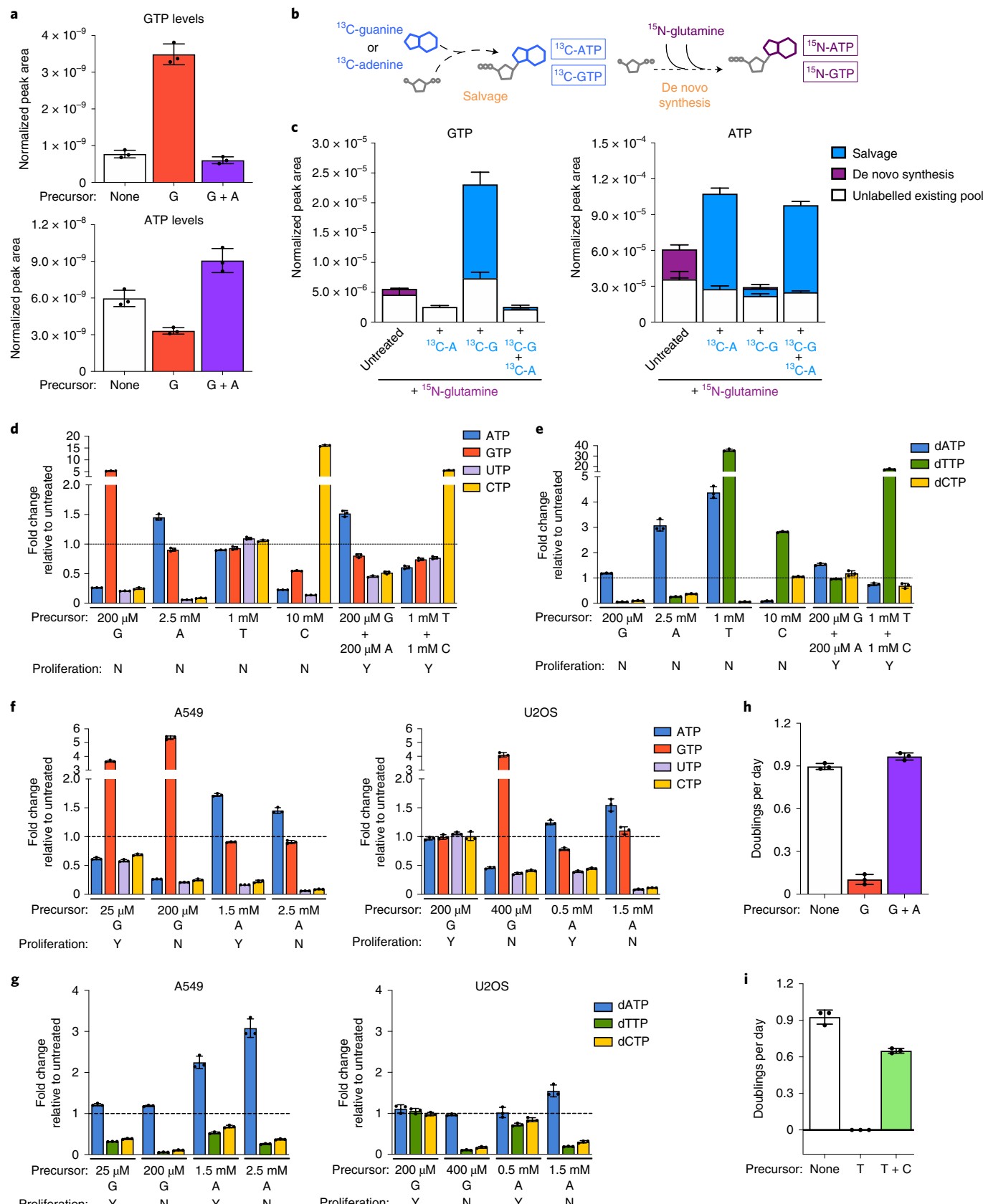

salvage disrupts relative levels of purines by increasing the ratio of G to A nucleotides. Notably, providing adenine together with guanine restored the balance of G and A nucleotides (Fig. 2a). To understand how providing guanine depletes intracellular A

nucleotides, we measured the contributions of salvage and de novo synthesis to intracellular purines. To assess de novo synthesis, we measured incorporation of amide-$^{15}$N-glutamine into purines, as the amide nitrogen of glutamine is incorporated during AMP and

**Fig. 2 | Nucleotide salvage leading to imbalanced nucleotide pools inhibits cell proliferation. a**, GTP and ATP levels in A549 cells cultured in standard conditions (none) or treated for 24 h with 200 μM guanine (G) with or without 200 μM adenine (A) as indicated. **b**, Schematic showing how stable isotope tracing was used to determine the source of intracellular purines. Salvage of [13]C-guanine or [13]C-adenine produces [13]C-labelled GTP and ATP. The [15]N label from amide-[15]N-glutamine is incorporated in de novo purine synthesis, producing [15]N-labelled ATP and GTP. **c**, Total levels and labelling of GTP and ATP in A549 cells cultured for 24 h in medium containing amide-[15]N-glutamine with or without 200 μM [13]C-guanine and/or [13]C-adenine as indicated. **d**, Fold change in the specified intracellular NTP levels in A549 cells cultured with the indicated concentrations of nucleotide precursors compared with those found in cells cultured in standard conditions. **e**, Fold change in the specified intracellular dNTP levels in A549 cells cultured with the indicated concentrations of nucleotide precursors compared with those found in cells cultured in standard conditions. **f**, Fold change in the specified intracellular NTP levels in A549 or U2OS cells cultured with the indicated concentrations of nucleotide precursors compared with those found in cells cultured in standard conditions. **g**, Fold change in the specified intracellular dNTP levels in A549 or U2OS cells cultured with the indicated concentrations of nucleotide precursors compared with those found in cells cultured in standard conditions. **h**, Proliferation rates of A549 cells cultured in standard conditions (none) or treated with 200 μM G with or without 200 μM A. **i**, Proliferation rates of A549 cells cultured in standard conditions (none) or treated with 1 mM T with or without 1 mM C. All nucleotide levels were measured using LCMS. Fold changes in nucleotide levels were calculated from absolute concentrations presented in Extended Data Fig. 2d–g. Data are presented as mean +/- SD of 3 biological replicates. Source numerical data are available in source data.

GMP synthesis. To assess salvage, we measured incorporation of [13]C-guanine and [13]C-adenine into purines (Fig. 2b). As expected, a subset of purines in untreated cells were [15]N-labelled, reflecting their production via de novo synthesis (Fig. 2c and Extended Data Fig. 2b). Providing [13]C-adenine increased levels of A nucleotides, the majority of which were [13]C-labelled and therefore derived from adenine salvage. Similarly, salvage of [13]C-guanine accounted for increased G nucleotide levels upon guanine supplementation. Moreover, providing either [13]C-adenine or [13]C-guanine eliminated the contribution of de novo synthesis to intracellular purines. This likely reflects known allosteric feedback regulation of purine synthesis enzymes: A and G nucleotides can inhibit ribose-5-phosphate pyrophosphokinase and glutamine phosphoribosyl pyrophosphate amidotransferase, which catalyse the initial steps of de novo purine synthesis[3,24,25]. Therefore, aberrantly high G nucleotides derived from guanine salvage can inhibit de novo synthesis of both G and A nucleotides (Extended Data Fig. 2c), resulting in A nucleotide depletion.

We hypothesized that analogous imbalances in nucleotide levels account for impaired proliferation upon salvage of other nucleotide precursors (Fig. 1b–f). To test this, we used liquid chromatography–mass spectrometry (LCMS) to quantify absolute intracellular nucleotide levels upon addition of A, G, T and C precursors. (Extended Data Fig. 2d,e). At concentrations that inhibit proliferation, each precursor increased intracellular concentrations of at least one nucleotide species and decreased intracellular concentrations of at least one other nucleotide species. Consistent with T being exclusive to the dNTP pool, T treatment caused altered dNTP levels but not NTP levels. Determining the fold change in levels of each intracellular nucleotide caused by A, G, T or C addition revealed that salvage of different precursors altered relative nucleotide levels in different ways (Fig. 2d,e). Thus, a change in any specific nucleotide species does not explain

decreased proliferation across these conditions. Rather, these data argue that cells are vulnerable to multiple different perturbations to the balance among nucleotide species. This led us to define nucleotide imbalance as a detrimental increase in one or more nucleotide species above normal levels along with a decrease in one or more other nucleotide species below normal levels. Importantly, this is distinct from depletion of purines, pyrimidines or all NTP or dNTP species.

To understand what degree of imbalance is needed to impair proliferation, we took advantage of differential sensitivity to nucleotide precursors across cell types. A549 cells are more sensitive than U2OS cells to G, but are less sensitive to A (Fig. 1b,f and Extended Data Fig. 1b). Comparing A549 and U2OS cells treated with concentrations of G or A that differentially impair proliferation revealed that similar magnitudes of change to nucleotide levels were detrimental to each cell type (Fig. 2f,g and Extended Data Fig. 2f,g). This suggests that differential sensitivity to each nucleotide may be attributable to varied transport or salvage activity across cell types and underscores that nucleotide imbalances are detrimental to proliferation. Indeed, providing adenine to re-establish purine balance restored proliferation of guanine-treated cells (Fig. 2d,e,h and Extended Data Fig. 2d,e,h). Providing cytidine also restored nucleotide balance and proliferation of thymidine-treated cells (Fig. 2d,e,i and Extended Data Fig. 2d,e).

Salvage of individual nucleotides altered intracellular levels of both NTPs and dNTPs. dGTP has the same molecular weight as ATP and similar chromatographic properties, and because ATP is much more abundant in cells, dGTP was not confidently distinguished by LCMS. Nevertheless, addition of each nucleotide precursor at concentrations that inhibit proliferation caused imbalances among dNTP species (Fig. 2e,g and Extended Data Fig. 2e,g), raising the possibility that imbalanced dNTPs play a role in impairing proliferation upon nucleotide precursor addition.

**Fig. 3 | Nucleotide imbalance impairs S phase progression. a**, Approach using flow cytometry to assess cell cycle phase by DNA content (as determined by propidium iodide staining) and EdU incorporation. **b**, Cell cycle distribution of A549 cells cultured with the indicated concentration of guanine (G) for 24 h. **c**, Cell cycle distribution of A549 cells treated with or without 200 μM G with or without 200 μM adenine (A) for 24 or 48 h. **d**, Approach to assess S phase progression. After pulsing cells with EdU, cell cycle progression of EdU-positive and EdU-negative populations was monitored. **e**, Cell cycle distribution of A549 cells pulsed with EdU, then cultured with or without 200 μM G for the indicated time. Percentage of total cells that are EdU-positive and in G1, S or G2/M phase is shown. **f**, mVenus-Gem1 fluorescent reporter to assess cell cycle dynamics in live cells. **g**, Representative images from live-cell imaging of A549 cells expressing mVenus-Gem1 cultured with or without 200 μM G (see also Supplementary Videos 1–3). **h**, Fraction of cells cultured with or without 200 μM G that began the experiment in G1 phase and entered S phase (assessed by live-cell imaging of A549 cells expressing mVenus-Gem1; 76 cells were analysed). **i**, Duration of S/G2 phase in cells cultured with or without 200 μM G (assessed by live-cell imaging of A549 cells expressing mVenus-Gem1; 115 cells were analysed). **j**, Cell cycle distribution of A549 cells synchronized in G2 phase with 4.5 μM RO-3306 for 18 h, then released from arrest and treated with standard culture media (untreated), 25 μM G (low G) or 200 μM G (high G) as indicated. **k**, dNTP levels in A549 cells 21 h after release from RO-3306 and subsequent treatment with or without low G or high G as indicated. dNTP levels in unsynchronized cells cultured with or without low G or high G for 24 h are also shown. dNTPs were measured using LCMS. Data are presented as mean ± s.d. of three biological replicates. Source numerical data are available in source data.

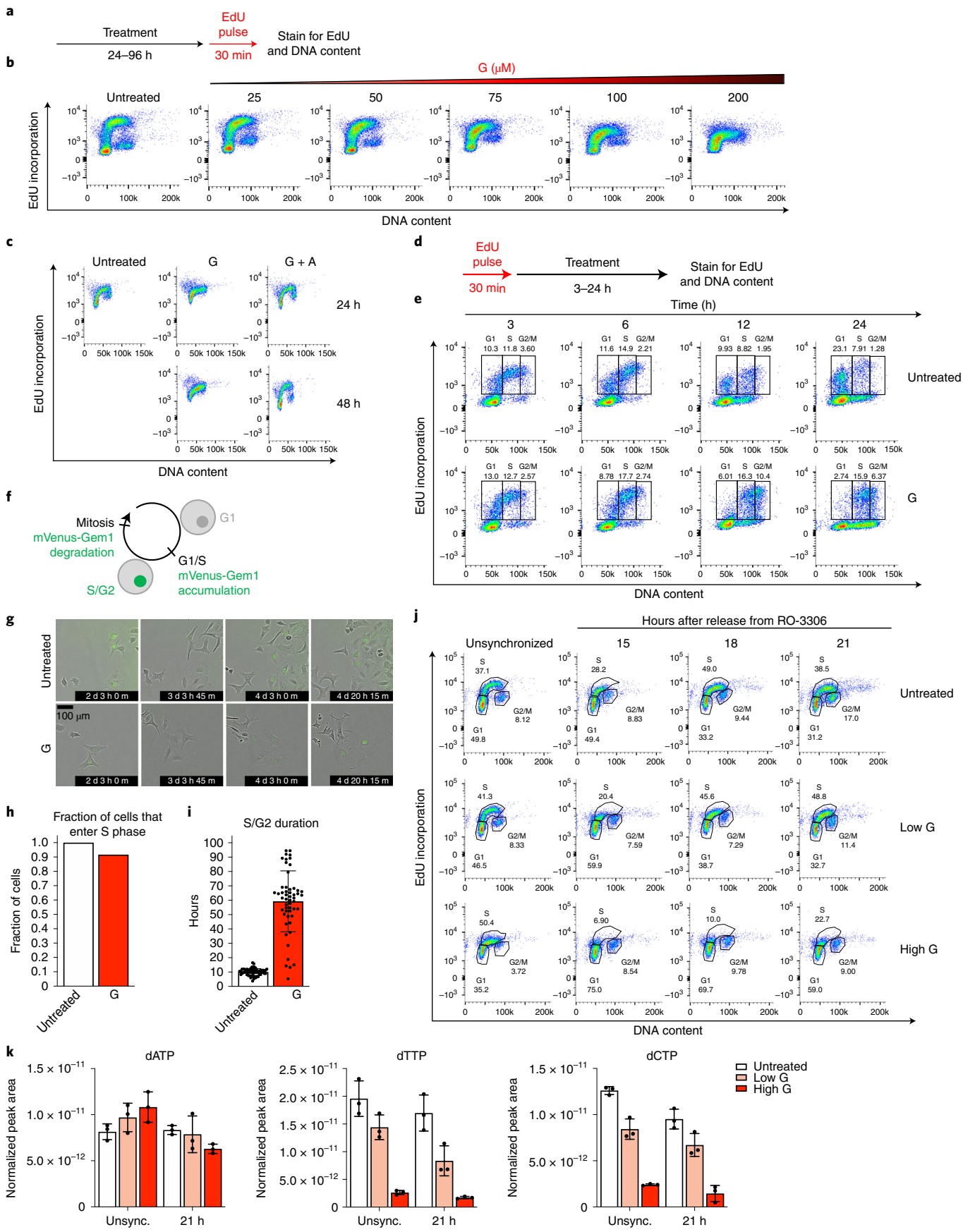

**Nucleotide imbalance slows S phase progression.** To test whether nucleotide imbalance impairs proliferation by impeding DNA replication during S phase, we monitored cell cycle progression following G treatment. We used flow cytometry to measure DNA content and incorporation of 5-ethynyl-2′-deoxyuridine (EdU) into DNA, which reflects active DNA replication (Fig. 3a). Untreated cells contain populations in all cell cycle phases; as reported in classic studies, serum starvation causes G1 arrest, while Taxol treatment causes G2/M arrest (Extended Data Fig. 3a,b)[26]. Consistent with its effect on proliferation (Fig. 1b), guanine treatment had a dose-dependent effect on cell cycle progression: increasing concentrations of guanine caused cells to accumulate in S phase, and at the highest concentration cells failed to incorporate EdU (Fig. 3b and Extended Data Fig. 3c). Similarly, guanine treatment for 96 h initially increased the population of cells in S phase and later prevented EdU incorporation (Extended Data Fig. 3d). Providing adenine together with guanine to restore nucleotide balance restored normal cell cycle distribution (Fig. 3c and Extended Data Fig. 3c). Imbalances caused by other nucleotide precursors also impaired S phase progression across cell types (Extended Data Fig. 3e,f). Treatment with LTX or BRQ to deplete all purines or pyrimidines, respectively, prevented EdU incorporation by 96 h but did not cause the same extent of S phase accumulation as guanine treatment (Extended Data Fig. 3d). Thus, while nucleotide imbalance impairs proliferation by slowing S phase progression, purine or pyrimidine depletion may inhibit proliferation in part through a different mechanism.

To more directly test whether nucleotide imbalance slows DNA replication, we pulsed cells with EdU to mark the population in S phase at $t = 0$, then monitored S phase progression. In untreated cells, the EdU-positive population progressed to 4 N DNA content and then back to 2 N DNA content, reflecting completion of S phase and return to G1 after cell division (Fig. 3d,e). This does not occur when cells are arrested in G1 or G2/M (Extended Data Fig. 3g). While untreated EdU-positive cells completed S phase and divided within 24 h, guanine-treated EdU-positive cells failed to divide by 24 h (Fig. 3e), suggesting that DNA replication is slowed. Further, a population of EdU-negative cells with intermediate DNA content accumulated during guanine treatment. Initially, EdU-negative cells with 2 N DNA content are in G1 phase. Thus, accumulation of EdU-negative cells at intermediate DNA content argues that cells enter S phase with nucleotide imbalance, but progression through S phase is impaired.

To define kinetics of S phase entry and duration, we performed live-cell imaging using a previously described fluorescent reporter[27] where mVenus is conjugated to a truncated form of geminin, whose degradation is cell cycle dependent. Cells expressing mVenus-Gem1 have fluorescent nuclei between the G1/S transition and mitosis, allowing for specific monitoring of S phase entry and quantification of S/G2 and G1 durations (Fig. 3f,g and Supplementary Videos 1–3). Almost all guanine-treated cells entered S phase, but subsequently

had much longer S/G2 duration than untreated cells (Fig. 3h,i). Guanine treatment also increased G1 duration in cells born after induction of nucleotide imbalance, though not to the same extent as S/G2 duration (Extended Data Fig. 3h).

We next tested whether dNTP imbalance persisted through S phase by synchronizing cells in G2 phase using the CDK1 inhibitor RO-3306 (ref. [28]) and then releasing cells into the following cell cycle. Importantly, this strategy does not directly perturb cell metabolism. Untreated cells entered S phase around 15 h after release and progressed to late S phase around 21 h after release, while high concentrations of guanine resulted in slower S phase progression (Fig. 3j). Guanine treatment caused dNTP imbalance 21 h after release (Fig. 3k), demonstrating that nucleotide imbalance perturbs dNTP availability during S phase. Together, these data suggest that cells lack a mechanism to prevent S phase entry with imbalanced nucleotides, leading to impaired DNA replication and S phase progression.

**Growth control pathways do not sense nucleotide imbalance.** As numerous growth signalling pathways regulate nucleotide metabolism, we asked whether these pathways decrease growth in coordination with decreased proliferation under nucleotide imbalance. Although mTORC1 responds to purine depletion[11,12], we found that mTORC1 signalling remains active despite nucleotide imbalance (Fig. 4a and Extended Data Fig. 4a,d). Activity of other major growth regulatory pathways, Akt and AMPK, also did not correlate with proliferation arrest under nucleotide imbalance (Extended Data Fig. 4b,c). Interestingly, decreased mTORC1 activity upon nucleotide depletion did not prevent continued growth in H1299 cells (Extended Data Fig. 4d,k). Additionally, guanine-treated cells grew aberrantly large, and incorporation of puromycin into nascent peptides showed that protein synthesis rates were unchanged (Fig. 4b,c). Indeed, guanine-treated cells synthesized protein in coordination with increasing cell volume, as measured by a YFP protein synthesis reporter[29,30] (Fig. 4d and Extended Data Fig. 4e). Thus, protein concentration and overall cell density are maintained despite a larger cell size (Fig. 4e,f). These data suggest that canonical growth signalling does not directly respond to nucleotide imbalance.

Other nucleotide imbalances also caused cells to grow aberrantly large in a dose-titratable manner that matched their anti-proliferative effects and was observed across cell types. (Fig. 4g and Extended Data Fig. 4f,g). Adding nucleobases/nucleosides at concentrations that do not affect proliferation did not change cell size (Extended Data Fig. 4h). Re-establishing nucleotide balance restored normal size in cells treated with G- or T-nucleotide precursors (Fig. 4h,i). Most other metabolic perturbations did not robustly increase cell size, with the exception of pyrimidine synthesis inhibition (Extended Data Fig. 4i). Pyrimidine depletion also did not inhibit mTORC1 signalling, as previously reported[11,12]. Thus, while metabolic state, growth and proliferation are normally

**Fig. 4 | Nucleotide imbalance causes continued cell growth without division. a**, Phosphorylation of ribosomal protein S6 and S6 kinase (S6K) in A549 cells cultured with or without 1 µM Torin 1, or 200 µM guanine (G), 1 µM lometrexol (LTX) or 1 µM brequinar (BRQ) for the indicated time. Levels of vinculin, total S6K and total S6 are also shown. **b**, Proliferation rate (left) and mean volume (right) of A549 cells cultured with or without 200 µM G. **c**, Global protein synthesis measured by puromycin incorporation into nascent peptides in A549 cells cultured with or without 200 µM G for 96 h. Cycloheximide treatment was used as a negative control. **d**, Cell number (left), mean volume (centre) and protein accumulation (right) in A549 cells treated with 200 µM G. Protein accumulation was determined using a YFP reporter (Extended Data Fig. 4e). **e**, Protein concentration in A549 cells cultured with or without 200 µM G, calculated by dividing total protein by cell number and volume. **f**, Density of A549 cells cultured with or without 200 µM G for 72 h, calculated by dividing cell mass by cell volume. **g**, Mean volume of A549 cells treated for 96 h with the indicated concentrations of G, thymidine (T), deoxyadenosine (dA), cytidine (C) or adenine (A). **h**, Mean volume of A549 cells cultured with or without 200 µM G with or without 200 µM A for 96 h. **i**, Mean volume of A549 cells cultured with or without 1 mM T with or without 1 mM C for 96 h. **j**, Proliferation rate and size of A549 cells cultured in conditions that perturb cell metabolism. Data are compiled from experiments shown in Figs. 1a–f, 2h,i and 4g–i, and Extended Data Fig. 4j. Conditions are grouped into signalling disruption (Torin treatment or serum withdrawal), amino acid limitation (leucine or arginine starvation), electron transport chain (ETC) inhibition (oligomycin or rotenone treatment), purine or pyrimidine depletion (using LTX or BRQ), or nucleotide imbalance. Data are presented as mean ± s.d. of three biological replicates. Source numerical data and unprocessed blots are available in source data.

tightly linked, cell growth is decoupled from proliferation following nucleotide imbalance (Fig. 4j).

The purine synthesis inhibitor LTX depletes both A and G nucleotides and inhibits proliferation (Extended Data Figs. 1a and 4j,k). In cells where LTX inhibits growth and mTORC1 activity, we asked whether supplementing purine-depleted cells with either adenine or guanine to cause purine imbalance decouples growth from proliferation. Adenine and guanine can reactivate mTORC1 in purine-depleted cells, but the time required for A versus G nucleotides to induce mTORC1 activity may be variable[11,12]. We found that both guanine and adenine could activate mTORC1 acutely following purine depletion and sustain signalling over longer time periods

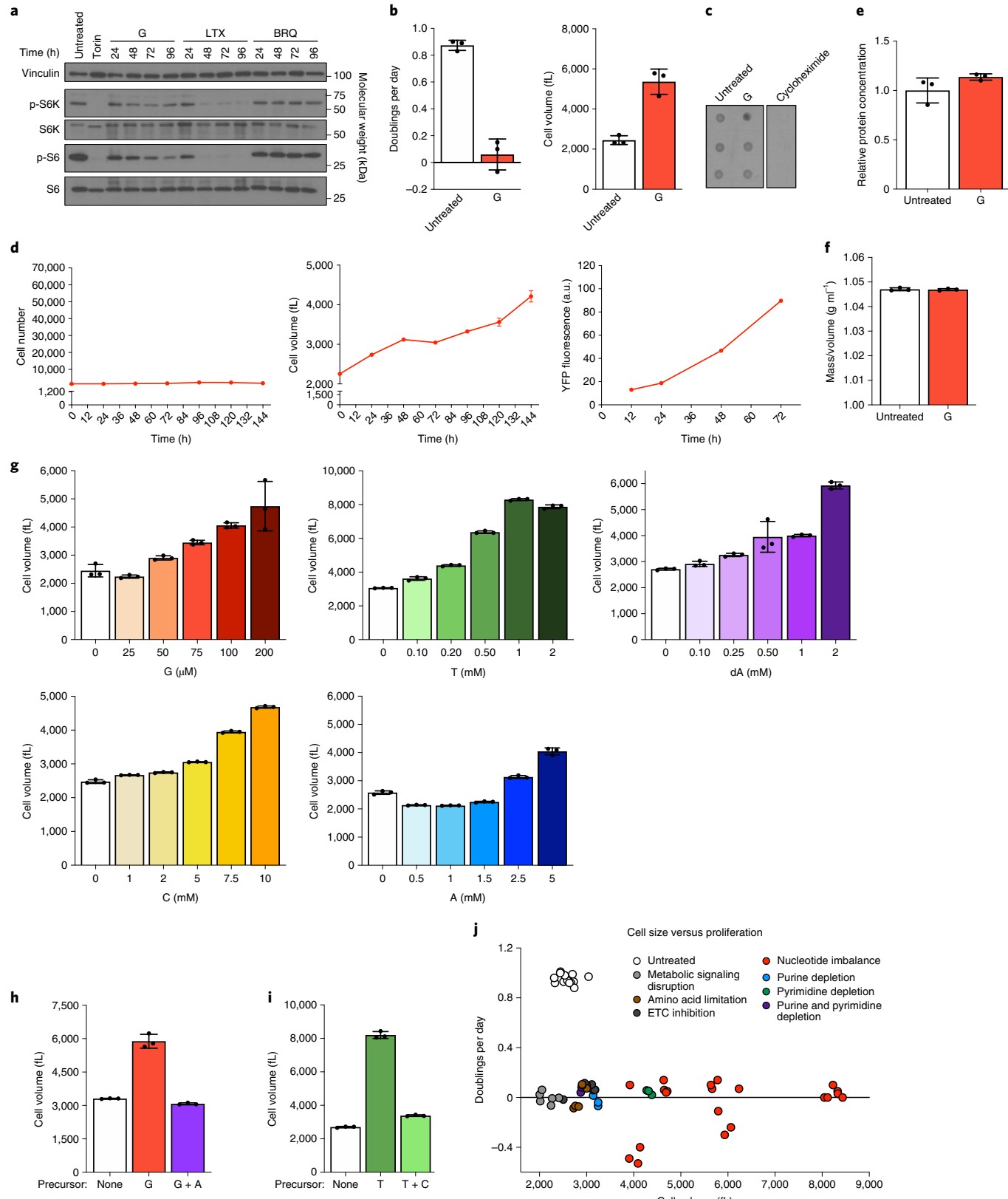

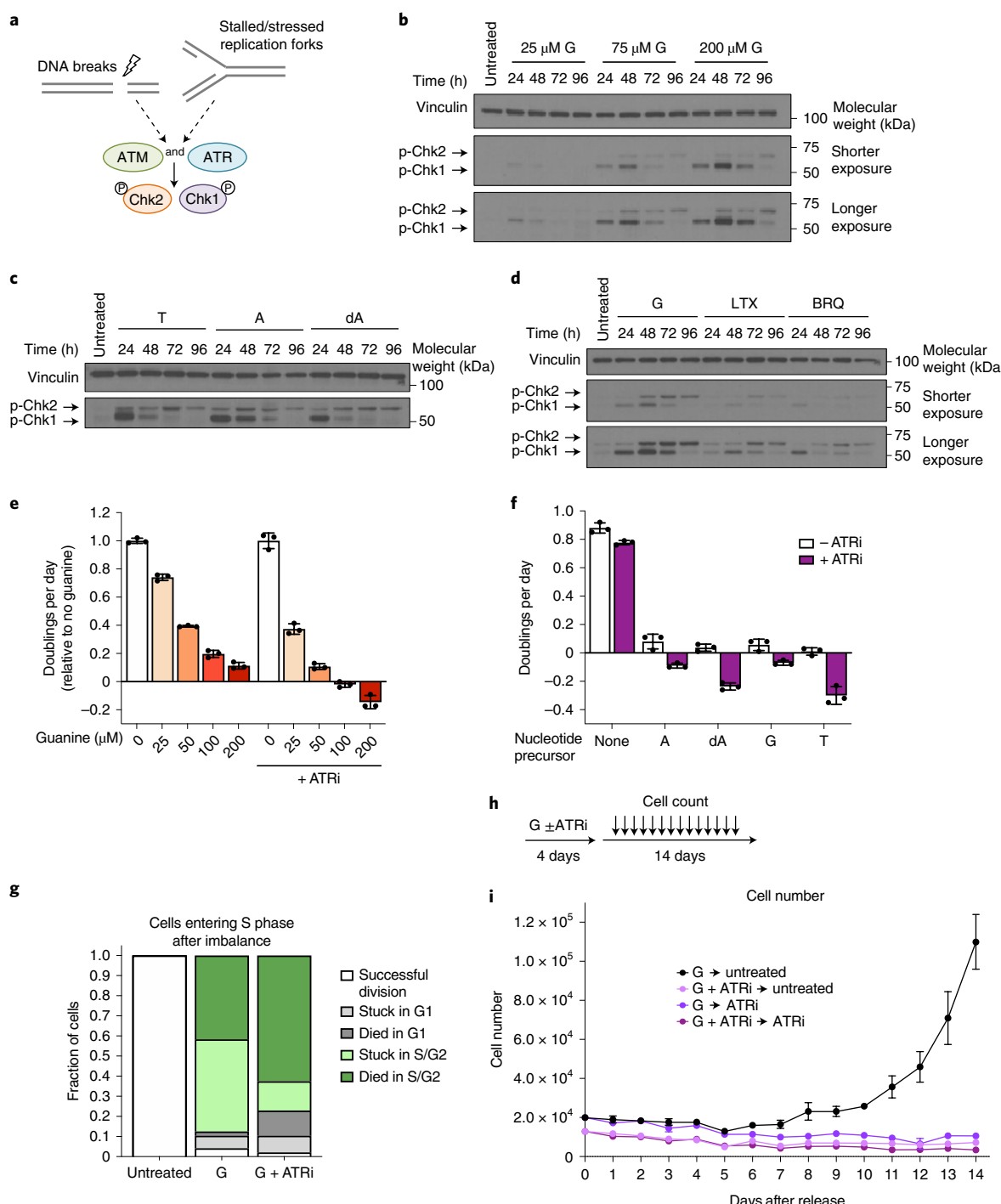

**Fig. 5 | Replication stress signalling promotes cell survival and recovery from nucleotide imbalance. a**, ATR and ATM kinases respond to replication stress and DNA damage. ATR and ATM phosphorylate Chk1 and Chk2, respectively. **b**, Phosphorylation of Chk1 and Chk2 in A549 cells treated for the indicated time with the indicated concentration of guanine (G). **c**, Phosphorylation of Chk1 and Chk2 in A549 cells treated for the indicated amount of time with 1 mM thymidine (T), 2.5 mM adenine (A) or 1.5 mM deoxyadenosine (dA). **d**, Phosphorylation of Chk1 and Chk2 in A549 cells treated for the indicated time with 200 μM G, 1 μM LTX or 1 μM BRQ. Levels of vinculin are also shown in all western blots as a loading control. **e**, Proliferation rates of A549 cells treated with the indicated concentration of guanine with or without 50 nM of the ATR kinase inhibitor AZ20 (ATRi) as indicated. **f**, Proliferation rates of A549 cells cultured with or without 2 mM A, 1.5 mM dA, 200 μM G or 1 mM T, with or without 50 nM ATRi as indicated. **g**, Cell fate of A549 cells expressing the mVenus-Gem1 reporter that were in G1 phase at the time of addition of 200 μM G with or without 50 nM ATRi, as assessed using live-cell imaging. The fate of cells in that were in G1 at the beginning of the experiment and were not exposed to excess G is also shown (untreated). In total, 124 cells were analysed. **h**, Approach to assess how cells recover from treatment with excess G. Cells were cultured in medium containing 200 μM G with or without 50 nM ATRi for 4 days. Medium was then changed to untreated medium or medium containing 50 nM ATRi, and cell number was determined every 24 h for 14 days thereafter. **i**, A549 cell number over time after release from treatment with G with or without ATRi treatment as outlined in **h**. Data are presented as mean ± s.d. of three biological replicates. Source numerical data and unprocessed blots are available in source data.

(Extended Data Fig. 4l,m). However, activation of growth signalling is not sufficient for proliferation: providing excess adenine or guanine did not restore proliferation (Extended Data Fig. 4n). Of note, low concentrations of adenine that do not induce nucleotide imbalance could rescue proliferation of LTX-treated cells. This may be explained by the ability of AMP deaminase to convert AMP to IMP, which can then be converted to GMP to potentially replenish both A and G nucleotides. Together, these data suggest that, while sufficient levels of either purine can restore growth, balanced levels of purines are required for proliferation. Further, while purine-depleted cells (with inactive mTORC1 signalling) accumulate in G1 phase, providing guanine to these cells caused S phase entry and subsequent S phase stalling (Extended Data Fig. 4o). We therefore hypothesized that mTORC1 activity is needed for S phase entry under nucleotide imbalance, and consistent with this, pharmacological inhibition of mTORC1 prevented guanine-treated cells from entering S phase (Extended Data Fig. 4o).

**Nucleotide imbalance activates replication stress signalling.** Impaired S phase progression suggests stalled DNA replication, and we therefore tested whether nucleotide imbalance causes DNA replication stress. The ATR and ATM kinases sense single-stranded DNA and DNA double-strand breaks, respectively, and their respective downstream targets, Chk1 and Chk2, are major DNA damage response (DDR) effectors (Fig. 5a)[31–33]. Guanine treatment caused robust phosphorylation of both Chk1 and Chk2, with higher concentrations of guanine that inhibit proliferation to a greater extent inducing a stronger signalling response (Fig. 5b). Interestingly, Chk1 was phosphorylated first within 24 h, followed by phosphorylation of Chk2 between 48 and 72 h. This may indicate that following purine imbalance, replication fork stalling first activates ATR, with later activation of ATM. Addition of adenine together with guanine prevented replication stress response induction (Extended Data Fig. 5a). Nucleotide imbalances induced by other precursors also activated ATR and ATM signalling, while as expected, using leucine deprivation to inhibit proliferation did not (Fig. 5c and Extended Data Fig. 5a–c). Inhibiting total purine or pyrimidine synthesis induced phosphorylation of Chk1 and Chk2 to a lesser extent than guanine treatment, consistent with fewer cells stalling in S phase in these conditions (Fig. 5d and Extended Data Fig. 3d).

Recent work demonstrated that pharmacological G nucleotide synthesis inhibition with the IMPDH inhibitor mycophenolic acid (MPA) can have dose-dependent effects: low-dose MPA increased p53 and p21 protein levels and caused cells to accumulate in G1 phase after 24 h, whereas high-dose MPA caused p21 degradation and increased the number of cells in S phase[34]. We tested whether guanine addition has similar dose-dependent effects on p53 and p21 levels. Consistent with its effects on ATR and ATM signalling, increasing concentrations of guanine increased p53 levels; however, higher guanine concentrations did not increase p21 degradation (Extended Data Fig. 5d). This suggests that nucleotide

synthesis inhibition and excess nucleotide salvage can have differing effects on cells.

The role of ATR and ATM in the cellular response to DNA damage has been extensively studied[31,32]; however, only a small fraction of guanine-treated cells exhibited minor increases in DNA damage at 24 h as measured by a comet tail assay (Extended Data Fig. 5e). At that time, the signalling response is already robust, indicating that replication stress-sensing pathways are activated under nucleotide imbalance without large amounts of DNA damage. Further, the failure of metabolic regulatory mechanisms to prevent S phase entry with imbalanced nucleotides suggests that replication stress sensing constitutes the major signalling response to nucleotide imbalance.

**ATR activity is required to survive nucleotide imbalance.** ATR and ATM activate downstream effectors that block cell cycle progression (Extended Data Fig. 6a)[31,33,35]. ATR-mediated cell cycle arrest might therefore explain why nucleotide imbalance prevents proliferation. If so, inhibiting ATR would allow cells to continue proliferating despite nucleotide imbalance. Using the ATR inhibitor AZ20 (ref. [36,37]), we found that, instead of restoring proliferation, ATR inhibition increased cell death following guanine treatment. (Fig. 5e and Extended Data Fig. 6b–d). ATR inhibition increased sensitivity to all nucleotide imbalances but did not increase sensitivity to purine or pyrimidine depletion (Fig. 5f and Extended Data Fig. 6e–g), consistent with a less robust induction of ATR signalling in these conditions. Together, these data suggest that replication stress signalling can be a protective mechanism to enable cell survival with nucleotide imbalance.

Live-cell imaging with the mVenus-Gem1 cell cycle reporter (Fig. 3f) showed that cells can successfully divide if they have partially completed S phase before guanine addition, though ATR inhibition caused some of these cells to arrest or die (Extended Data Fig. 6h). This is consistent with the expected kinetics of nucleotide imbalance: because these cells turn over purine nucleotide pools in approximately 24 h (ref. [18]), inhibition of A nucleotide synthesis following excess guanine salvage (Fig. 2c and Extended Data Fig. 2c) would not immediately deplete A nucleotides. Thus, cells that have already partially replicated their DNA can likely complete replication before purine balance is drastically changed. In contrast, cells in G1 at the time of guanine supplementation that enter S phase with imbalanced nucleotides are unable to divide, and instead either arrest or die after entering S phase (Fig. 5g). ATR inhibition caused most cells to die upon entering S phase with imbalanced purines, suggesting that ATR is critical for preventing S phase catastrophe under nucleotide imbalance. Interestingly, daughter cells born later after induction of nucleotide imbalance and ATR inhibition were more likely to become stalled in G1 (Extended Data Fig. 6i). Replication stress in mother cells can affect G1 length and lead to quiescence in daughter cells[38], potentially implying that an inadequate replication

**Fig. 6 | ATR signalling promotes dNTP availability during unperturbed S phases. a**, Cell cycle distribution of A549 cells corresponding to western blots in **b**. Cells were treated with 9 μM RO-3306 for 18 h to arrest cells in G2 phase, then released for the indicated time. **b**, Phosphorylation of Chk1 and Chk2 in A549 cells treated with 9 μM RO-3306 for 18 h to arrest cells in G2 phase, then released for the indicated time. Levels of vinculin are also shown. **c**, Cell fate assessed using live-cell imaging of A549 mother cells expressing mVenus-Gem1 that were in G1 phase at the time of 50 nM AZ20 (ATRi) addition. The fate of mother cells in G1 not exposed to ATRi is also shown (untreated). Fifty-five cells were analysed. **d**, Cell fate of A549 daughter cells expressing the mVenus-Gem1 reporter born to mother cells either in S/G2 phase (left) or G1 phase (right) at the time of 50 nM ATRi addition. Mother cells that were in S/G2 phase when ATRi was added went through a partial S phase with ATR inhibited, whereas mother cells that were in G1 phase went through a full S phase with ATR inhibited. The fate of daughter cells not exposed to ATRi is also shown (untreated). For left and right graphs, 104 and 107 cells were analysed, respectively. **e**, dNTP levels in A549 cells synchronized in G2 phase by treating with 4.5 μM RO-3306 for 18 h, then released from RO-3306 and treated with DMSO or 50 nM ATRi for the indicated times. Unsynchronized cells (unsync.) were treated with DMSO or 50 nM ATRi for 24 h as indicated. **f**, NTP levels in A549 cells synchronized in G2 phase by treating with 4.5 μM RO-3306 for 18 h, then released and treated with DMSO or 50 nM ATRi for the indicated times. Unsync. cells were treated with DMSO or 50 nM ATRi for 24 h as indicated. Nucleotide levels were measured using LCMS. Data are presented as mean ± s.d. of three biological replicates. Source numerical data and unprocessed blots are available in source data.

stress response under nucleotide imbalance can result in DNA damage that is inherited by daughter cells and affects their proliferative potential.

We next asked whether cells can recover from nucleotide imbalance. A subset of cells showed evidence of senescence induction at prolonged timepoints following guanine treatment, consistent with

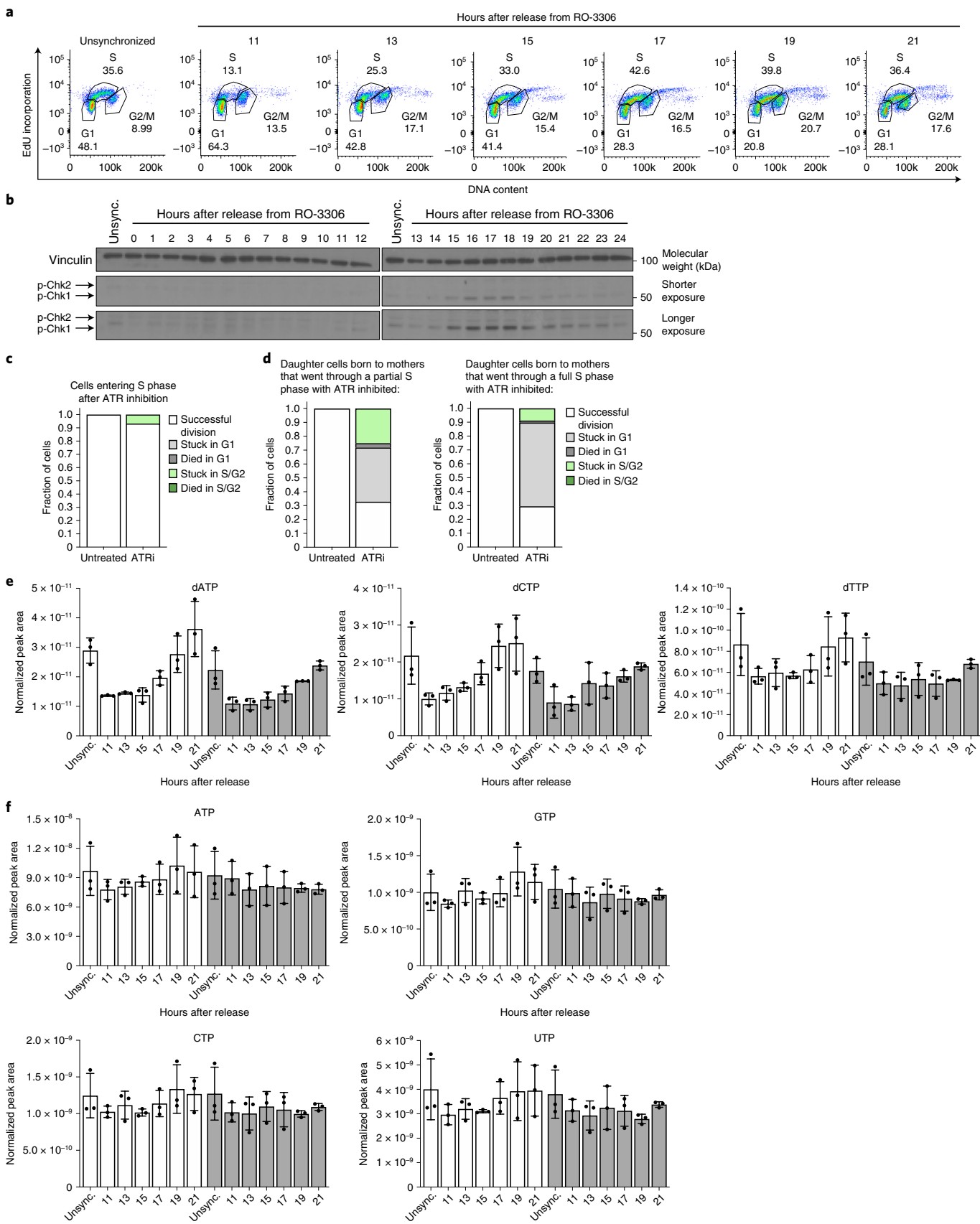

replication stress being a driver of cellular senescence (Extended Data Fig. 6j). However, on a population level, cells resumed proliferation at a normal size after release from nucleotide imbalance. Strikingly, ATR inhibition either concurrent with or after guanine treatment prevented recovery (Fig. 5h,i and Extended Data Fig. 6k), arguing that failure to activate a replication stress response under nucleotide imbalance causes irreversible damage to cells. ATR inhibition can allow inappropriate firing of late origins of DNA replication, and we reasoned that this may force cells to continue through S phase at a pace that is incompatible with dNTP imbalance. However, ATR inhibition during guanine treatment did not appear to accelerate cell cycle progression (Extended Data Fig. 6l), suggesting that specific dNTPs themselves may become limiting as substrates for replication following nucleotide imbalance.

**ATR activity supports dNTP levels in unperturbed S phases.** The fact that cells enter S phase despite imbalanced nucleotides suggests that cells do not monitor nucleotide balance before S phase entry, and replication stress signalling may be important for adjusting nucleotide availability during normal proliferation. Indeed, ATR activity has been observed in unperturbed cell cycles[35,39,40], and the yeast ATR homologue Mec1 is activated as a result of low dNTPs at the onset of S phase[5]. As ATR promotes accumulation of the RNR subunit RRM2 during S phase[41], we tested whether replication stress signalling allows mammalian cells to maintain sufficient dNTPs for DNA replication. We first monitored replication stress sensing during normal S phases by synchronizing cells in G2 phase and releasing cells into the following cell cycle. ATR was activated as most cells entered early S phase, and was attenuated as most cells progressed through late S phase (Fig. 6a,b and Extended Data Fig. 7a).

We next monitored how ATR activity impacts cell cycle progression using live-cell imaging of mVenus-Gem1-expressing cells. Both G1 and S/G2 duration increased upon ATR inhibition, with G1 duration increasing to a greater extent (Extended Data Fig. 7b). Mother cells treated with ATR inhibitor successfully completed their current cell cycle (Fig. 6c and Extended Data Fig. 7c). However, the majority of daughter cells born to ATR-inhibited mother cells had slow cell cycle progression and became stalled in G1 (Fig. 6d). The likelihood of G1 stalling was greater for daughter cells whose mothers underwent an entire S phase with ATR inhibited compared with daughters whose mothers only experienced ATR inhibition for the latter part of S phase. These results are consistent with the idea that inability to activate ATR signalling during an otherwise unperturbed S phase may result in DNA damage that is inherited by daughter cells and causes them to stall in G1 (ref. [38]).

We next asked whether ATR modulates dNTP levels during S phase. While levels of other metabolites, including amino acids and NTPs, were relatively constant throughout the cell cycle, dNTP levels were lower as most cells entered S phase and increased as cells progressed through S phase (Fig. 6e,f and Extended Data Fig. 7d,e). Thus, ATR activation correlates with low dNTP levels upon S phase entry. These data are consistent with cells entering S phase with insufficient dNTPs for replication fork progression, leading to replication stress. As downstream effectors of ATR can activate nucleotide synthesis enzymes[42], we asked whether ATR activity promotes dNTP production during normal S phases. Indeed, ATR inhibition attenuated the increase in dNTP levels over the course of S phase (Fig. 6e and Extended Data Fig. 7d). Together, these data suggest that cells do not sense nucleotide levels in preparation for S phase, but may instead rely on replication stress signalling to modulate dNTP availability for genome replication.

## Discussion

The failure of growth regulatory pathways to sense nucleotide imbalance results in continued cell growth and biomass production. That cells continue to produce protein argues that RNA synthesis is largely not impaired by nucleotide imbalance, despite ribosomal RNA accounting for the majority of nucleic acid biomass in cells. One possibility is that even when NTP levels are imbalanced, the availability of each NTP species is still sufficient for RNA synthesis: NTP levels are generally at least an order of magnitude higher than dNTP levels[14]. For example, while guanine treatment decreased intracellular adenylate nucleotide pools, baseline ATP levels are high relative to dNTPs, and may not become limiting for RNA production.

Activation of replication stress signalling could promote survival under nucleotide imbalance in part by activating enzymes to stabilize replication forks and replenish dNTPs while preventing additional origin of replication firing. ATR is activated and important for the proper sequence of origin firing during unperturbed S phases[35,39,40]. Further, ATR effectors can activate RNR to promote dNTP synthesis[42,43]. Our finding that ATR activity is needed to increase dNTP availability during unperturbed S phases is consistent with a role for replication stress signalling in responding to nucleotide levels and suggests that ATR may be important for allowing cells to adapt to fluctuating nucleotide levels encountered during normal cell divisions. It is possible that dNTP imbalance occurs stochastically as dNTPs are rapidly consumed during genome replication; a stochastic decrease in different dNTPs in individual cells would lead to a measured reduction in all dNTPs in a bulk population. In budding yeast, the ATR homologue Mec1 is activated in early S phase downstream of initially low dNTP pools[5], indicating that this metabolic role may be conserved across eukaryotes.

Many cancers harbour mutations in DNA damage response (DDR) pathways[44]. As replication stress signalling is essential for survival during nucleotide imbalance, DDR-deficient tumours could be especially sensitive to perturbed nucleotide balance. Dysregulated expression of nucleotide salvage and catabolism enzymes may also render certain cancers vulnerable to imbalance. The dNTP-degrading enzyme SAMHD1 protects against cytotoxic dGTP buildup upon deoxyguanosine supplementation: SAMHD1-deficient tumour cells are sensitive to dGTP accumulation caused by deoxyguanosine supplementation and purine nucleoside phosphorylase inhibition[45,46]. Nucleotide imbalance is also implicated in non-cancer disease settings. Purine nucleoside phosphorylase deficiency and adenosine deaminase deficiency exhibit aberrant accumulation of dGTP and dATP, respectively[47–49] and lead to severe immunodeficiencies with insufficient T- and B-cell proliferation. Ameliorating nucleotide imbalance may improve fitness of these cells.

Cells can recover from nucleotide imbalance and resume proliferation, but senescence induction occurs in a subset of cells at longer intervals following release from imbalance. What determines whether a given cell becomes senescent is not clear. Replication stress is known to contribute to senescence[50,51], and one possibility is that the capability of an individual cell to resolve replication stress determines whether it escapes senescence following nucleotide imbalance. Indeed, we found that cells with inhibited replication stress signalling fail to proliferate following nucleotide imbalance and continue to grow excessively large, consistent with classic descriptions of cellular senescence.

The fact that mean cell volume returns to normal upon recovery from nucleotide imbalance suggests that cells have an established 'target' size and that cell populations can return to that average size. Division is likely necessary for cell volume reduction, consistent with the observation that cell number begins to increase before mean size decreases. In addition, cell growth plateaus with prolonged arrest due to nucleotide imbalance, implying that mechanism(s) also exist to halt biomass production despite initial uncoupling of cell growth and division. Nevertheless, it is unclear why cells continue to grow upon release from nucleotide imbalance despite already being aberrantly large; this suggests that growth is not initially tightly controlled with respect to target cell size.

More generally, this study shows that cell growth and division can be decoupled downstream of nucleotide imbalances that might occur in response to fluctuating nutrient levels. Replication stress signalling modulates nucleotide availability during normal proliferation and protects against fluctuations in nucleotide levels, but larger environmental changes that affect nucleotide balance increase the risk of genomic damage, raising the possibility that nucleotide imbalance-induced replication stress plays additional roles in cell physiology or function.

## Online content

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

## Methods

**Cell lines and cell culture.** All cell lines were cultured in Dulbecco's modified Eagle's medium (DMEM) (Gibco) supplemented with 10% heat-inactivated foetal bovine serum (FBS) at 37 °C with 5% $CO_2$. Cell lines were obtained from ATCC (catalogue numbers A549 – CCL-185; H1299 – CRL-5803; 143B – CRL-8303; U2OS – HTB-96; MDA-MB-468 – HTB-132; RPE-1 – CRL-4000); A9 cells were a gift from the B. Manning laboratory. All cell lines regularly tested negative for mycoplasma. To generate cells with stable transgene expression, Lenti-X 293 T cells at 75% confluency were transfected using X-tremeGENE 9 DNA transfection reagent (Sigma). The lentiviral plasmids used were pRSV-Rev (Addgene #12253), pMDLg/pRRE (Addgene #12251) and pMDG2.G (Addgene #12259) from Didier Trono. For the YFP-DHFR reporter, the donor expression plasmid pLJM1-FLAG-YFP-DHFR[30] was used. For the mVenus-Gem1 reporter, the donor expression plasmid pLenti-puro-mVenus-Gem1 was used. After 48 h, lentivirus was collected by removing the culture medium from the Lenti-X 293 T cells and passing it through a 0.45 μm filter. The target cell lines at 50–60% confluency were then infected using 3 ml virus with polybrene reagent (Sigma). After 24 h, virus was removed and cells were allowed to recover in virus-free medium for 24 h. Selection was then initiated using puromycin at a concentration of 2 μg ml⁻¹.

**Proliferation rates and cell size measurements.** All cell lines were plated in six-well plates in DMEM with 10% FBS at a concentration of 20,000 cells per well with the exception of MDA-MB-468 cells, which were plated at a concentration of 40,000 cells per well. The number of cells seeded for each cell line allowed for exponential growth over the course of the assay. The following day, one six-well plate of each cell line was counted to determine the initial number of cells at the time of treatment. Cells were washed three times with PBS, and 4 ml of treatment medium was added. Treatment medium was made with 10% dialysed FBS. Medium lacking specific amino acids was made from DMEM without pyruvate or amino acids supplemented with an amino acid mix containing DMEM concentrations of amino acids without arginine, leucine or serine. Arginine, leucine or serine was added back to the medium as needed. After 4 days of treatment, final cell counts were measured using a Multisizer 3 Coulter Counter (Beckman Coulter). The following formula was used to calculate proliferation rate:

Doublings per day = [log₂(final day 4 cell count /initial day 0 cell count)]/4 days

Cell size measurements were taken using a Multisizer 3 Coulter Counter at the same time as cell counts, after 4 days of treatment.

**Cell density.** The average cell density in a population was measured by comparing Multisizer 3 Coulter Counter (Beckman Coulter)-based cell volume measurements and suspended microchannel resonator (SMR)-based buoyant mass measurements. The resulting relative density values were converted to absolute density values by also measuring the density of the measurement solution (culture medium) using SMR and by calibrating the SMR measurements using polystyrene beads of known volume[52,53]. Population average densities were measured after 3 days of nucleobase treatment, because longer nucleobase treatments resulted in cells that were too large for the SMR microchannels.

**Protein synthesis: puromycin incorporation.** In total, 25,000 cells were plated in 6-cm plates. The following day, cells were washed three times with PBS and 5 ml of treatment medium containing 10% dialysed FBS was added for the specified amount of time. To perform the puromycin pulse, cells were kept at 37 °C and puromycin was added to the culture medium at 10 μg ml⁻¹ for exactly 1 min. Cells were washed once in ice-cold PBS, and the plates were flash frozen in liquid nitrogen and subsequently stored at −80 °C. Then, 10 μg ml⁻¹ cycloheximide was added to a negative control plate 30 min before the puromycin pulse. Protein lysates were prepared using ice-cold RIPA buffer with protease inhibitor. To perform dot blots, samples were normalized for protein concentration and 2 μl of lysate was dotted onto a 0.22-μm nitrocellulose membrane. Membranes were blocked with 5% milk for 1 h, washed and incubated at 4 °C overnight with the following primary antibodies in 5% BSA in TBST: puromycin (Sigma, 1:25,000) and vinculin (Cell Signaling Technology, 1:1,000). The following day, membranes were washed three times with TBST on a rocker for 10 min. Secondary antibodies were applied for 60 min. Anti-rabbit (Cell Signaling Technology) secondary antibody was used at a dilution of 1:5,000, and anti-mouse (Cell Signaling Technology) secondary antibody was used at a dilution of 1:10,000. Membranes were then washed again three times with TBST for 10 min, and signal was detected with enhanced chemiluminescent (ECL) substrate using film.

**Protein synthesis: YFP reporter.** To assess protein production and accumulation over time, a previously described protein synthesis reporter was employed[29,30]. In this reporter system, YFP is fused to an engineered unstable *Escherichia coli* dihydrofolate reductase (DHFR). YFP is rapidly degraded, and accumulates only in the presence of the ligand trimethoprim (TMP), which stabilizes the DHFR domain. Accumulation of fluorescence over time in the presence of TMP therefore reflects protein synthesis rate of the YFP reporter. To monitor YFP production, 80,000 cells were seeded in six-well plates and allowed to adhere overnight. The following day, cells were washed three times with PBS and 4 ml of treatment medium containing 10 μM TMP was added. After the indicated amount of time, cells were trypsinized, pelleted and resuspended in PBS. YFP fluorescence was measured using flow cytometry.

**Protein concentration.** Protein concentration was calculated by dividing total protein content by cell number and cell volume for each sample. A BCA assay was used to measure total protein content, as compared with a standard curve. At the same time, cell number and volume were measured in a parallel sample using a Multisizer 3 Coulter Counter (Beckman Coulter).

**Cell cycle analysis by flow cytometry.** In total, 500,000 cells were plated in 10-cm plates and incubated overnight to allow cells to adhere. The following day, cells were washed three times with PBS and 10 ml of treatment medium with 10% dialysed FBS was added for the desired amount of time. EdU was spiked into cell culture plates at 10 μM for exactly 30 min before fixing. To fix cells, each plate was trypsinized, pelleted and washed twice with PBS. Cells were resuspended in 500 μl ice-cold PBS, and 5 ml ice-cold ethanol was added dropwise to each sample while vortexing in order to obtain a single-cell suspension. Fixed cells were stored at 4 °C until being processed by flow cytometry (no longer than 4 days).

Cells were stained for EdU using the Click-iT EdU Pacific Blue kit (Invitrogen) according to manufacturer instructions. After staining for 30 minutes, cells were stained with propidium iodide. Cells were pelleted and washed with 1% BSA in PBS, then resuspended in 800 μl 1% BSA in PBS. Then, 200 μl propidium iodide/ RNAseA staining solution was added to each sample and cells were stained for at least 45 min at 4 °C protected from light. Samples were then passed through a 0.35-μm filter into flow cytometry tubes (Falcon). Samples were run on a BD FACSCanto II Cell Analyzer, and 10,000 events were recorded for each sample. BD FACSDiva software was used to collect data. FlowJo software was used to analyse the data.

**Comet assay.** A total of 500,000 cells were plated in 10-cm plates and incubated overnight to allow cells to adhere. The following day, cells were washed three times with PBS and 10 ml of treatment medium containing 10% dialysed FBS was added. After 24 h of treatment, cells were trypsinized, pelleted and resuspended in ice-cold PBS at a concentration of $1 \times 10^5$ cells ml⁻¹. Sample preparation, electrophoresis, staining and microscopy were then performed using a CometAssay Kit (Trevigen) according to the manufacturer's instructions. Percent of DNA in comet tail was quantified using ImageJ with OpenComet software[54].

**Western blots.** In total, 10⁶ cells were plated in 10-cm plates and incubated overnight to allow cells to adhere. The following day, cells were washed three times with PBS and 10 ml of treatment medium containing 10% dialysed FBS was added. After culturing cells for the indicated time in treatment medium, protein lysates were prepared by rapidly placing cells on ice and washing with ice-cold PBS, then lysing cells in ice-cold RIPA buffer containing cOmplete protease inhibitor (Roche) and phosphatase inhibitor cocktail (Sigma-Aldrich). Lysed cells were vortexed at 4 °C for 10 min and then centrifuged at maximum speed at 4 °C for 10 min. Protein lysate supernatant was removed and stored at −80 °C. Proteins were separated using SDS–PAGE (12% acrylamide gels) and transferred to a nitrocellulose membrane using a standard wet transfer method. Membranes were blocked for 60 min using 5% BSA in TBST. Membranes were incubated in primary antibody overnight at 4 °C. The following primary antibodies were used at a dilution of 1:1,000: vinculin (Cell Signaling Technology #4650), phospho-ribosomal protein S6 Ser 235/236 (CST #4858), ribosomal protein S6 (CST #2217), phospho-p70 S6 kinase Thr389 (CST #9205), p70 S6 kinase (CST #9202), phospho-Akt (CST #4060), Akt (CST #9272), phospho-AMPK (CST #2535), AMPK (CST #2532), phospho-Chk1 Ser345 (CST #2348) (for Extended Data Fig. 6f only), phospho-Chk1 Ser345 (CST #2341) (for all other blots showing p-Chk1), phospho-Chk2 Thr68 (CST,2197), p53 (CST #9282) and p21 (CST, #2947). Antibodies were diluted in 5% BSA in TBST. The following day, membranes were washed three times with TBST on a rocker for 10 min. Secondary antibodies were applied for 60 min. Anti-rabbit (CST #7074) secondary antibody was used at a dilution of 1:5,000, and anti-mouse (CST #7076) was used at a dilution of 1:10,000. Membranes were then washed again three times with TBST for 10 min, and signal was detected with ECL using film (except for the blot shown in Extended Data Fig. 6f, which was imaged using a BioRad ChemiDoc System).

**LCMS analysis.** A total of 100,000 cells were plated in six-well plates in DMEM with 10% FBS and incubated overnight. The following day, cells were washed three times with PBS and 4 ml of treatment medium was added. All treatment medium was made with 10% dialysed FBS. After the indicated time period, polar metabolites were extracted from cells: plates were placed on ice, cells were washed with ice-cold blood bank saline and 500 μl of ice-cold 80% methanol in water with 250 nM ¹³C/¹⁵N-labelled amino acid standards (MSK-A2-1.2: Cambridge Isotope Laboratories, Inc.) was added to each well. Cells were scraped, each sample was vortexed for 10 min at 4 °C, and then centrifuged at maximum speed for 10 min at 4 °C. Samples were dried under nitrogen gas and resuspended in 25 μl of a 50/50 acetonitrile/water mixture. Metabolites were measured using a Dionex UltiMate 3000 ultrahigh-performance liquid chromatography system connected to a Q Exactive benchtop Orbitrap mass spectrometer, equipped with an Ion Max

source and a HESI II probe (Thermo Fisher Scientific). Samples were separated by chromatography by injecting 2–10 μl of sample on a SeQuant ZIC-pHILIC Polymeric column (2.1 × 150 mm 5 μM, EMD Millipore). Flow rate was set to 150 μl min⁻¹, temperatures were set to 25 °C for column compartment and 4 °C for autosampler sample tray. Mobile Phase A consisted of 20 mM ammonium carbonate, 0.1% ammonium hydroxide. Mobile Phase B was 100% acetonitrile. The mobile phase gradient (%B) was set in the following protocol: 0–20 min, linear gradient from 80% to 20% B; 20–20.5 min, linear gradient from 20% to 80% B; 20.5–28 min, hold at 80% B. Mobile phase was introduced into the ionization source set to the following parameters: sheath gas, 40; auxiliary gas, 15; sweep gas, 1; spray voltage, −3.1 kV; capillary temperature, 275 °C; S-lens RF level, 40; probe temperature, 350 °C. Metabolites were monitored in full-scan, polarity-switching mode. An additional narrow range full scan (220–700 $m/z$) in negative mode only was included to enhance nucleotide detection. The resolution was set at 70,000, the AGC target at 1,000,000 and the maximum injection time at 20 ms. Relative quantitation of metabolites was performed with XCalibur QuanBrowser 2.2 (Thermo Fisher Scientific) using a 5 ppm mass tolerance and referencing an in-house retention time library of chemical standards. Metabolite measurements were normalized to the internal ¹³C/¹⁵N-labelled amino acid standard and to cell number.

For absolute quantification of intracellular nucleotides, 100,000 cells were plated in six-well plates in DMEM with 10% FBS and incubated overnight. The following day, cells were washed three times with PBS and 4 ml of treatment medium was added. All treatment medium was made with 10% dialysed FBS. Cells were cultured in treatment medium for 24 h before polar metabolites were extracted and analysed as described above. In addition to 250 nM ¹³C/¹⁵N-labelled amino acid standards (MSK-A2-1.2, Cambridge Isotope Laboratories), the 80% methanol in water used to extract metabolites contained ¹³C/¹⁵N-labelled nucleotides to enable absolute quantification. The concentration of labelled nucleotide standards used in the extraction solvent was determined by first establishing standard curves of the labelled nucleotides. Concentrations that were close to the reported physiological ranges[14] and within the linear range of detection were chosen to spike into the extraction mix. For each sample, the final amounts of each standard (in picomoles) were as follows: dATP, 28.8; dGTP, 6.24; dCTP, 34.8; dTTP, 44.4; ATP, 3,782.4; GTP, 561.6; CTP, 333.6; UTP, 680.4.

**Stable isotope tracing to assess nucleotide synthesis and salvage.** In total, 100,000 cells were plated in six-well plates in 2 ml DMEM with 10% FBS and incubated overnight. The following day, cells were washed three times with PBS and 4 ml of treatment medium was added. All treatment medium was made with 10% dialysed FBS and 4 mM ¹⁵N-amide-glutamine. Then, 200 μM ¹³C-guanine or ¹³C-adenine was added to the treatment medium as indicated. Cells were cultured in treatment medium for 24 h before polar metabolites were extracted and analysed as described in 'LCMS analysis'.

**Live-cell imaging with cell cycle reporter.** Live-cell tracking of A549 cells expressing the mVenus-Gem1 reporter was carried out using the IncuCyte live-cell imaging system (Sartorius). The cells were plated at a density of 40,000 cells per well on six-well plates, and the 4-day long treatments were started 24 h after plating. Imaging was performed every 45 min using 10× objective and 150 ms exposure time for the FITC channel. Cell tracking was carried out manually using the IncuCyte software. Only cells that were mVenus-Gem1 positive at some point during the 24-h period before chemical treatment were tracked. Manual tracking recorded timings of cell cycle state change and/or cell death, as detected on the basis of cell morphology. The tracking of each cell lineage lasted until the division of second cell generation or until the end of the 4-day drug treatment. If a cell remained arrested in a specific cell cycle state until the conclusion of the experiment, the duration of that cell cycle state was calculated to end at the conclusion of the experiment. Consequently, for a small fraction of cells, the durations of some cell cycle phases are underestimated.

**Cell death measurements.** A total of 500,000 cells were plated in 10-cm plates and incubated overnight to allow cells to adhere. The following day, cells were washed three times with PBS and 10 ml of treatment medium with 10% dialysed FBS was added for 96 h. Cells were collected and stained with Live-or-Dye viability dye (Biotium) according to the manufacturer's instructions. Samples were then passed through a 0.35-μm filter into flow cytometry tubes (Falcon) and run on a BD FACSCanto II Cell Analyzer with 10,000 events recorded for each sample.

**Cell cycle synchronization.** In total, 150,000 cells were plated in 6-cm plates (for matched protein lysates and flow cytometry-based cell cycle analysis) or six-well plates (for matched LCMS-based metabolite measurements and flow cytometry-based cell cycle analysis) in DMEM with 10% FBS. The following day, cells were washed three times with PBS and medium was replaced with medium containing RO-3306 (Selleckchem), for which 4.5–9.0 μM RO-3306 was used, with the concentration for different lots of RO-3306 adjusted to obtain optimal synchronization for experiments. After treating cells for 18 h with RO-3306 (or DMSO for unsynchronized controls), cells were released from cell cycle arrest by washing three times with PBS and replacing the medium with untreated medium.

Where relevant, medium containing either 50 nM AZ20 (or DMSO as vehicle) or guanine was added at the time of release from RO-3306. Cells were collected at the indicated timepoints after release from arrest. For each experiment, parallel samples for each timepoint were analysed by flow cytometry to assess cell cycle distribution.

**SA-β-galactosidase assay.** A total of 50,000 cells were plated in six-well plates in DMEM with 10% FBS. The following day, cells were washed twice with PBS and medium was replaced with 2 ml DMEM with treatment medium. To change medium for recovery, cells were washed twice with PBS and medium was replaced. At the indicated time, cells were washed twice with PBS and fixed for 10 min with 1× fixative solution at room temperature. Cells were then washed twice again with PBS and stained for 48 h at 37 °C using the Senescence Beta-Galactosidase Staining Kit (CST #9860) and imaged on a Nikon Eclipse Ti.

**Statistics and reproducibility.** No data were excluded from the analyses. All assays were performed in triplicate as specified in figure legends, and representative experiments (such as western blots) shown in Figs. 3g, 4a, 5b–d and 6b and Extended Data Figs. 4a–d,l,m, 5a–d and 6b,f,j were repeated three times. Sample sizes were based on general practices in the field. No statistical method was used to pre-determine sample size. Randomization was not applicable to the study design, and the experiments were not amenable to blind allocation.

**Reporting summary.** Further information on research design is available in the Nature Research Reporting Summary linked to this article.

## Data availability

The raw numerical source data associated with experiments presented in this study are presented in the corresponding source data files for each figure. Unprocessed western blots are presented in the corresponding source data files for each figure. All other data supporting the findings of this study are available from the corresponding author upon reasonable request. Source data are provided with this paper.

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

## Acknowledgements

We thank members of the Vander Heiden lab for helpful discussions and K. Sapp for constructs to express the mVenus-Gem1 reporter. The A9 cells were a gift from the B. Manning laboratory. We also thank the Koch Institute Swanson Biotechnology Center Flow Cytometry Facility. This research was supported by the National Cancer Institute of the NIH under award number F31CA236036 (F.F.D.) and award numbers R35CA242379, R01CA201276 and P30CA14051 (M.G.V.H.), by the National Heart, Lung, and Blood Institute of the NIH under award number F30HL156404 (B.T.D.) and by the National Institute for General Medical Sciences of the NIH under award number T32GM007753 (B.T.D.). T.P.M was supported by the Wellcome Trust grant 110275/Z/15/Z. M.G.V.H. also acknowledges support from a Faculty Scholar grant from the Howard Hughes Medical Institute, SU2C, a division of the Entertainment Industry Foundation, the Lustgarten Foundation, the MIT Center for Precision Cancer Medicine and the Ludwig Center at MIT.

## Author contributions

Conceptualization, F.F.D. and M.G.V.H.; investigation, F.F.D., T.P.M., R.E., C.S.N., A.M.D., B.T.D. and C.A.L.; writing—original draft, F.F.D.; writing—review and editing, F.F.D., T.P.M., R.E., C.S.N., A.M.D., B.T.D., S.R.M., C.A.L. and M.G.V.H.; supervision, M.G.V.H.; funding acquisition, F.F.D. and M.G.V.H.

## Competing interests

The authors are aware of no direct conflicts with the topic of the study; however, F.F.D. and M.G.V.H. are included on a patent application regarding impact of nucleotide imbalance on cell state. S.R.M. declares he is a co-founder of Affinity Biosensors and Travera, and M.G.V.H. declares he is an advisory board member for Agios Pharmaceuticals, Aeglea Biotherapeutics, Faeth Therapeutics, Drioa Ventures and iTeos Therapeutics, and a co-founder of Auron Therapeutics. The other authors declare no competing interests.

## Additional information

**Extended data** is available for this paper at https://doi.org/10.1038/s41556-022-00965-1.

**Correspondence and requests for materials** should be addressed to Matthew G. Vander Heiden.

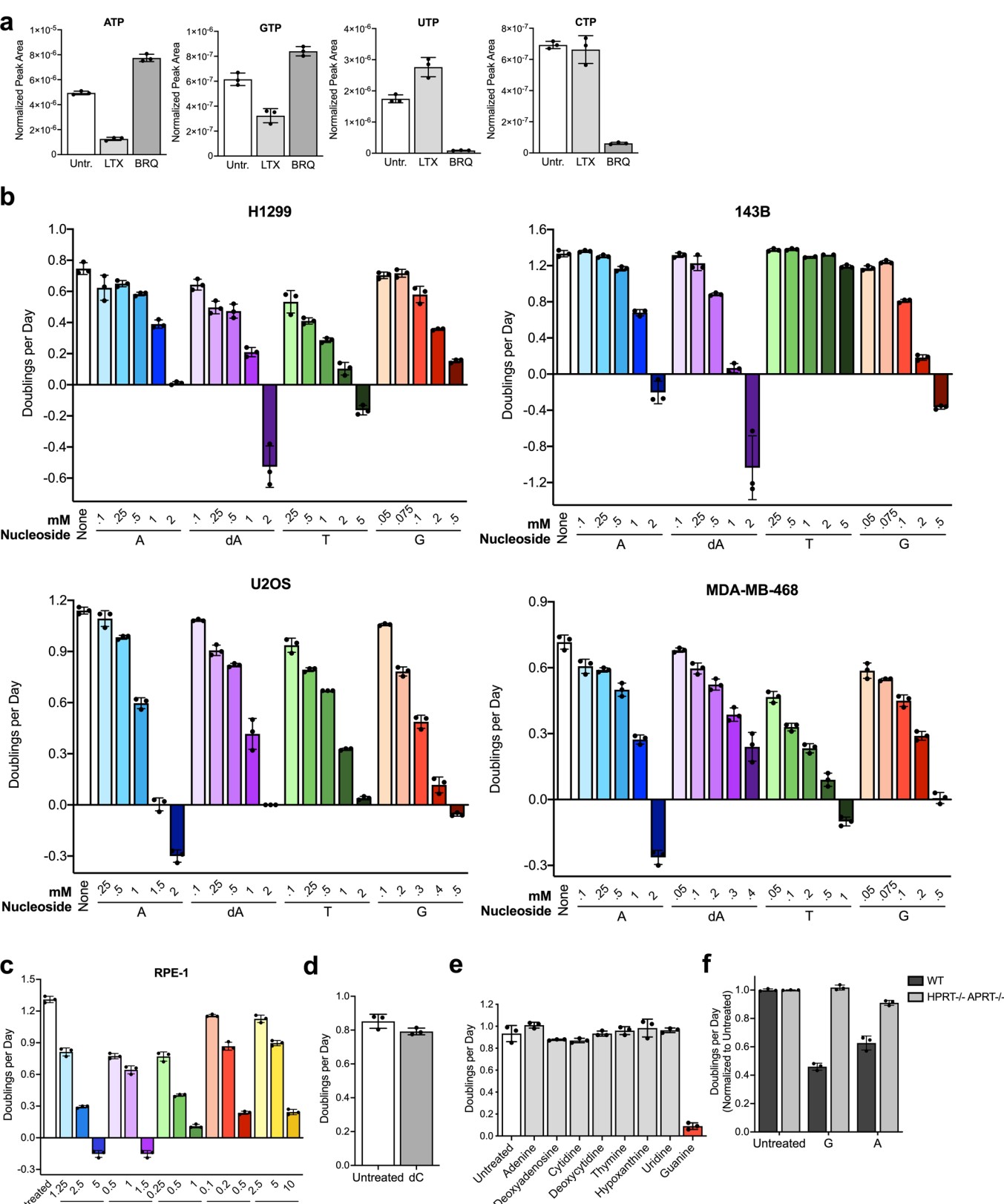

**Extended Data Fig. 1 | See next page for caption.**

**Extended Data Fig. 1 | Salvage of different nucleobases and nucleosides can inhibit proliferation in different cell types. a**, Intracellular nucleotide levels in A549 cells cultured in standard conditions (Untr.) or treated with 1 μM lometrexol (LTX) or 1 μM brequinar (BRQ) as indicated. **b**, Proliferation rates of the indicated cells in standard culture conditions (None) or treated with the indicated concentrations of adenine (A), deoxyadenosine (dA), thymidine (T), or guanine (G). Of note, 143B cells are deficient in thymidine kinase, and therefore cannot salvage thymidine to produce dTMP. **c**, Proliferation rates of RPE-1 cells in standard culture (Untreated) or treated with the indicated concentrations of A, dA, T, G, or cytidine (C). **d**, Proliferation rates of A549 cells in standard culture conditions (Untreated) or treated with 14 mM deoxycytidine (dC). **e**, Proliferation rates of A549 cells in standard culture conditions (Untreated) or treated with 200 μM of the indicated nucleobase/nucleoside. **f**, Normalized proliferation rates of A9 cells that are wild type (WT) or deficient (HPRT-/- APRT-/-) for hypoxanthine-guanine phosphoribosyltransferase and adenine phosphoribosyltransferase in standard culture conditions (Untreated) or treated with 200 μM G or A. Data are presented as mean +/- SD of 3 biological replicates. Source numerical data are available in source data.

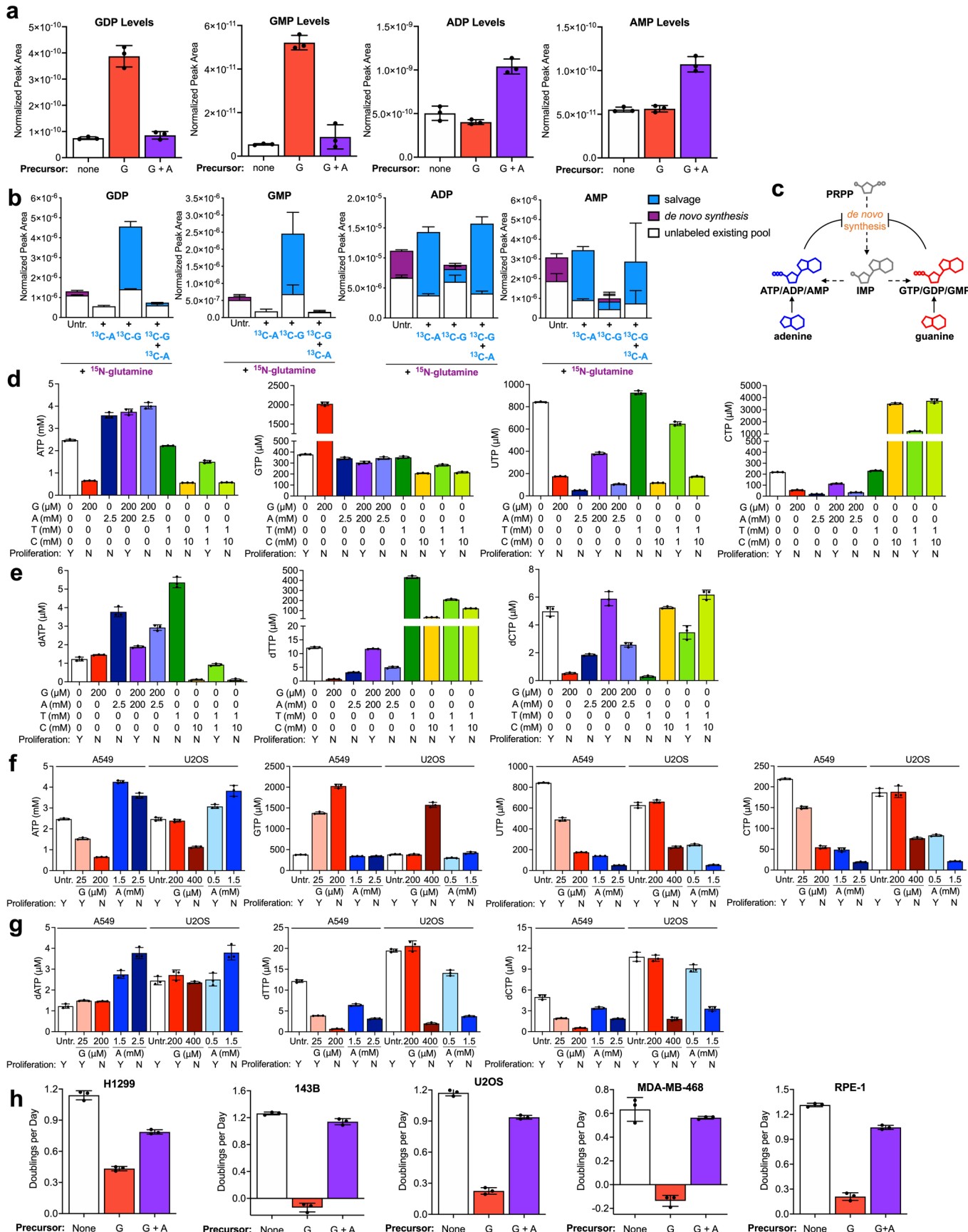

**Extended Data Fig. 2 | See next page for caption.**

**Extended Data Fig. 2 | Salvage of excess nucleotide precursors can induce nucleotide imbalance and impair proliferation. a**, Levels of the indicated nucleotides in A549 cells cultured in standard conditions (none) or treated for 24 hours with 200 μM guanine (G) with or without 200 μM adenine (A) as indicated. **b**, Total levels and labeling of the indicated nucleotides in A549 cells cultured for 24 hours in media containing $^{15}$N-amide-glutamine with or without 200 μM $^{13}$C-guanine ($^{13}$C-G) and/or $^{13}$C-adenine ($^{13}$C-A) as indicated. **c**, Diagram showing feedback regulation of purine synthesis. Adenylate and guanylate purines can allosterically inhibit enzymes involved in *de novo* purine synthesis. PRPP, phosphoribosyl pyrophosphate; IMP, inosine monophosphate. **d**, Absolute quantification of intracellular NTPs in A549 cells treated with the indicated concentrations of G, A, thymidine (T), or cytidine (C). Also indicated is whether supplementing culture media with that concentration of G, A, T and/or C does (Y) or does not (N) allow cell proliferation. **e**, Absolute quantification of intracellular dNTPs in A549 cells treated with the indicated concentrations of G, A, T, or C. Also indicated is whether supplementing culture media with that concentration of G, A, T and/or C does (Y) or does not (N) allow cell proliferation. **f**, Absolute quantification of intracellular NTPs in A549 cells and U2OS cells treated with the indicated concentrations of G or A. Also indicated is whether supplementing culture media with that concentration of G or A does (Y) or does not (N) allow cell proliferation. The data from A549 Untreated (Untr.), 200 μM G, and 2.5 mM A samples are the same as shown in panel d. **g**, Absolute quantification of intracellular dNTPs in A549 cells and U2OS cells treated with the indicated concentrations of G or A. Also indicated is whether supplementing culture media with that concentration of G or A does (Y) or does not (N) allow cell proliferation. The data from A549 Untr., 200 μM G, and 2.5 mM A samples are the same as shown in panel e. **h**, Proliferation rates of H1299, 143B, U2OS, MDA-MB-468, and RPE-1 cells in standard culture conditions (None) or treated with 200 μM G with or without 200 μM A, or 500 μM G with or without 500 μM A for RPE-1 cells, as indicated. All nucleotide levels were measured using LCMS. Data are presented as mean +/- SD of 3 biological replicates. Source numerical data are available in source data.

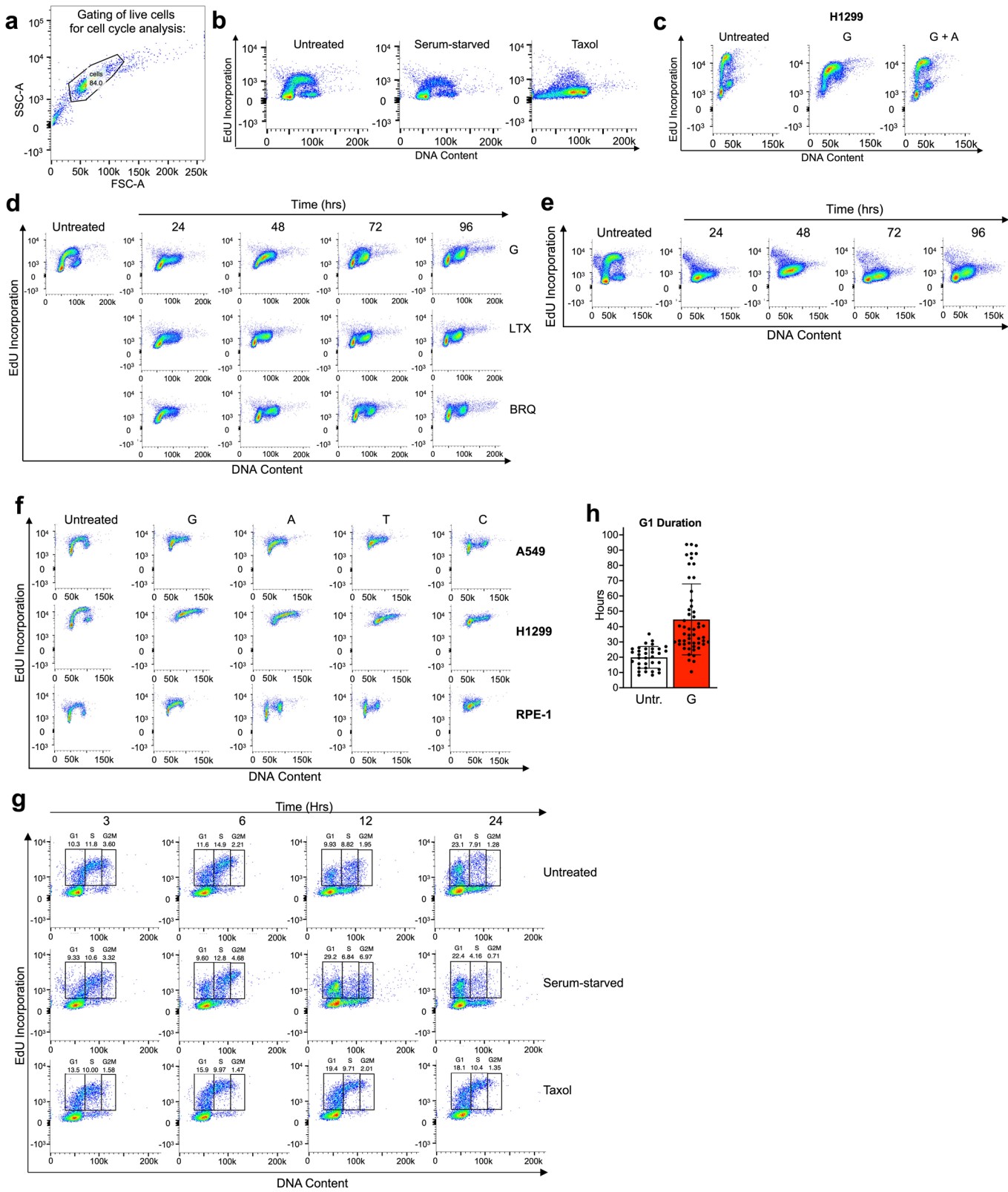

**Extended Data Fig. 3 | See next page for caption.**

**Extended Data Fig. 3 | Nucleotide imbalance and depletion differentially alter cell cycle progression. a**, Gating strategy for live cells based on forward scatter (FSC) and side scatter (SSC). **b**, Cell cycle distribution as assessed by propidium iodide staining and EdU incorporation of A549 cells cultured for 24 hours in standard media (Untreated), in media lacking FBS (Serum-starved), or with 10 μM Taxol. Cells were pulsed with EdU for 30 minutes after each treatment and analyzed as outlined in Fig. 3a. **c**, Cell cycle distribution of H1299 cells cultured for 24 hours in standard conditions (Untreated) or treated with 500 μM guanine (G) with or without 500 μM adenine (A) as indicated. Cells were pulsed with EdU for 30 minutes after each treatment and analyzed as outlined in Fig. 3a. **d**, Cell cycle distribution of A549 cells cultured in standard conditions (Untreated) or treated for the indicated amount of time with 200 μM G, 1 μM lometrexol (LTX), or 1 μM brequinar (BRQ) as indicated. Cells were pulsed with EdU for 30 minutes after each treatment and analyzed as outlined in Fig. 3a. **e**, Cell cycle distribution of A549 cells cultured in standard conditions (Untreated) or treated with 1 mM thymidine for the indicated amount of time. As EdU is a thymidine analog, thymidine supplementation is expected to blunt EdU incorporation. Cells were pulsed with EdU for 30 minutes after each treatment and analyzed as outlined in Fig. 3a. **f**, Cell cycle distribution of the indicated cells cultured in standard conditions (Untreated) or treated with the following concentrations of nucleotide precursors: A549 - 200 μM G, 1 mM T, 2.5 mM A, or 10 mM C; H1299 - 500 μM G, 1 mM T, 2.5 mM A, or 10 mM C; RPE-1 - 500 μM G, 1 mM T, 2.5 mM A, or 10 mM C. Cells were pulsed with EdU for 30 minutes after each treatment and analyzed as outlined in Fig. 3a. **g**, Cell cycle distribution of A549 cells pulsed with EdU (as outlined in Fig. 3d) and then cultured for the indicated amount of time in standard conditions (Untreated), in media lacking FBS (Serum-starved), or with 10 μM Taxol as indicated. The percentage of total cells that are EdU-positive and in G1, S, or G2/M phase is also shown. The Untreated samples shown are from the same experiment shown in Fig. 3e. **h**, Duration of G1 phase in A549 cells expressing an mVenus-Gem1 reporter (see Fig. 3f) cultured in standard conditions (Untr.) or with 200 μM G as assessed using live-cell imaging. 87 cells were analyzed. Data are presented as mean +/- SD. Source numerical data are available in source data.

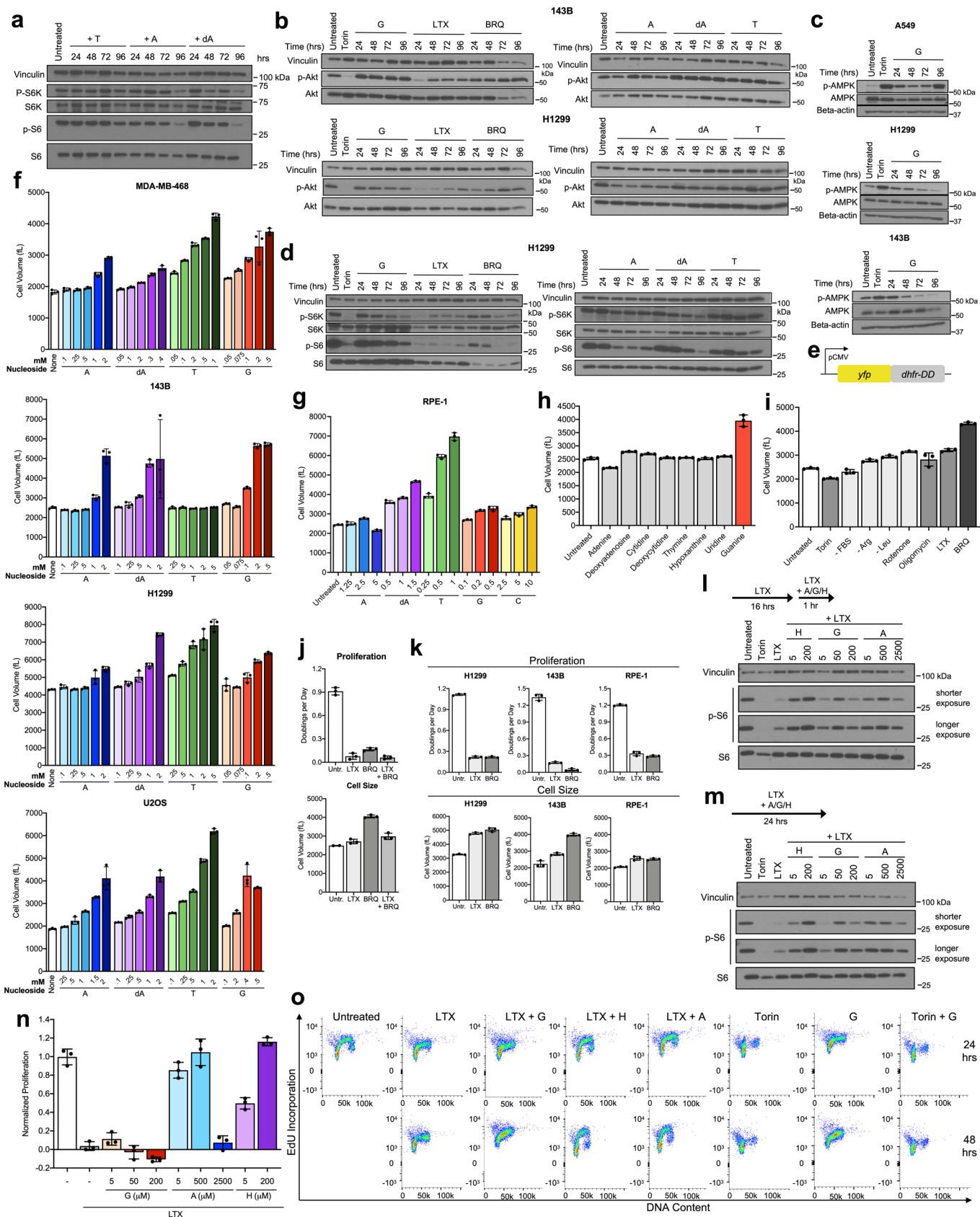

**Extended Data Fig. 4 | See next page for caption.**

**Extended Data Fig. 4 | Under nucleotide imbalance, continued cell growth and S phase entry do not correlate with changes in growth signaling. a**, Western blot assessing phosphorylation of ribosomal protein S6 and S6 kinase (S6K) in A549 cells cultured in standard conditions (Untreated) or treated with 1 mM thymidine (T), 2.5 mM adenine (A), or 1.5 mM deoxyadenosine (dA) for the indicated time. Levels of vinculin, total S6K, and total S6 are also shown as controls. **b**, Western blots assessing phosphorylation of Akt in 143B cells (top) and H1299 cells (bottom) cultured in standard conditions (Untreated) or treated with 1 μM Torin, 200 μM guanine (G), 1 μM lometrexol (LTX), 1 μM brequinar (BRQ), 1 mM T, 2.5 mM A, or 1.5 mM dA for the indicated time. Levels of vinculin and total Akt are also shown as controls. **c**, Western blots assessing phosphorylation of AMPK in A549, H1299, or 143B cells cultured in standard conditions (Untreated) or treated with 1 μM Torin or 200 μM G for the indicated time. Levels of vinculin and total AMPK are also shown as controls. **d**, Western blots assessing phosphorylation of ribosomal protein S6 and S6K in H1299 cells cultured in standard conditions (Untreated) or treated with 1 μM Torin, 200 μM G, 1 μM LTX, 1 μM BRQ, 1 mM T, 2.5 mM A, or 1.5 mM dA for the indicated time. Levels of vinculin, total S6K, and total S6 are also shown as controls. **e**, Schematic of a protein synthesis reporter construct where a CMV promoter drives expression of YFP fused to an engineered unstable *E. coli* dihydrofolate reductase that acts as a degron (*dhfr-DD*). **f**, Mean volume of the indicated cells cultured in standard conditions (None) or treated for 96 hours with the indicated concentration of A, dA, T, or G. 143B cells are deficient for thymidine kinase, and therefore cannot salvage thymidine to produce dTMP. **g**, Mean volume of RPE-1 cells cultured in standard conditions (None) or treated for 96 hours with the indicated concentration of A, dA, T, G, or cytidine (C). **h**, Mean volume of A549 cells cultured in standard conditions (Untreated) or treated with 200 μM of the indicated nucleobase/nucleoside for 96 hours. **i**, Mean volume of A549 cells cultured for 96 hours in standard culture conditions (Untreated), or with 1 μM Torin1, without serum (-FBS), without arginine (-Arg), without leucine (-Leu), with 100 nM rotenone, with 5 nM oligomycin, with 1 μM LTX, or with 1 μM BRQ as indicated. **j**, Proliferation rate (top) and mean volume (bottom) of A549 cells cultured in standard conditions (Untr.) or treated for 96 hours with 1 μM LTX, 1 μM BRQ, or both LTX and BRQ as indicated. **k**, Proliferation rate (top) and mean volume (bottom) of H1299, 143B, and RPE-1 cells cultured in standard conditions (Untr.) or treated for 96 hours with 1 μM LTX or 1 μM BRQ as indicated. **l**, Phosphorylation of ribosomal protein S6 in A549 cells cultured for 16 hours in standard conditions (Untreated) or with 1 μM LTX, then supplemented for 1 hour with the indicated concentrations in μM of hypoxanthine (H), G, or A. Levels of vinculin and total S6 are also shown as controls. **m**, Phosphorylation of ribosomal protein S6 in A549 cells cultured for 24 hours in standard conditions (Untreated) or treated with 1 μM LTX with or without the indicated concentrations in μM of H, G, or A. Levels of vinculin and total S6 are also shown as controls. **n**, Proliferation rates of A549 cells cultured in media with or without 1 μM LTX, and with or without the indicated concentrations of G, A, or H. **o**, Cell cycle distribution of A549 cells cultured in standard conditions (Untreated), or with 1 μM LTX with or without 200 μM G, 200 μM H, or 200 μM A as indicated, or with 10 μM Torin, 200 μM G, or both Torin and G for the indicated time. Cells were pulsed with EdU for 30 minutes after each treatment and then analyzed as outlined in Fig. 3a. Data are presented as mean ± SD of 3 biological replicates. Source numerical data and unprocessed blots are available in source data.

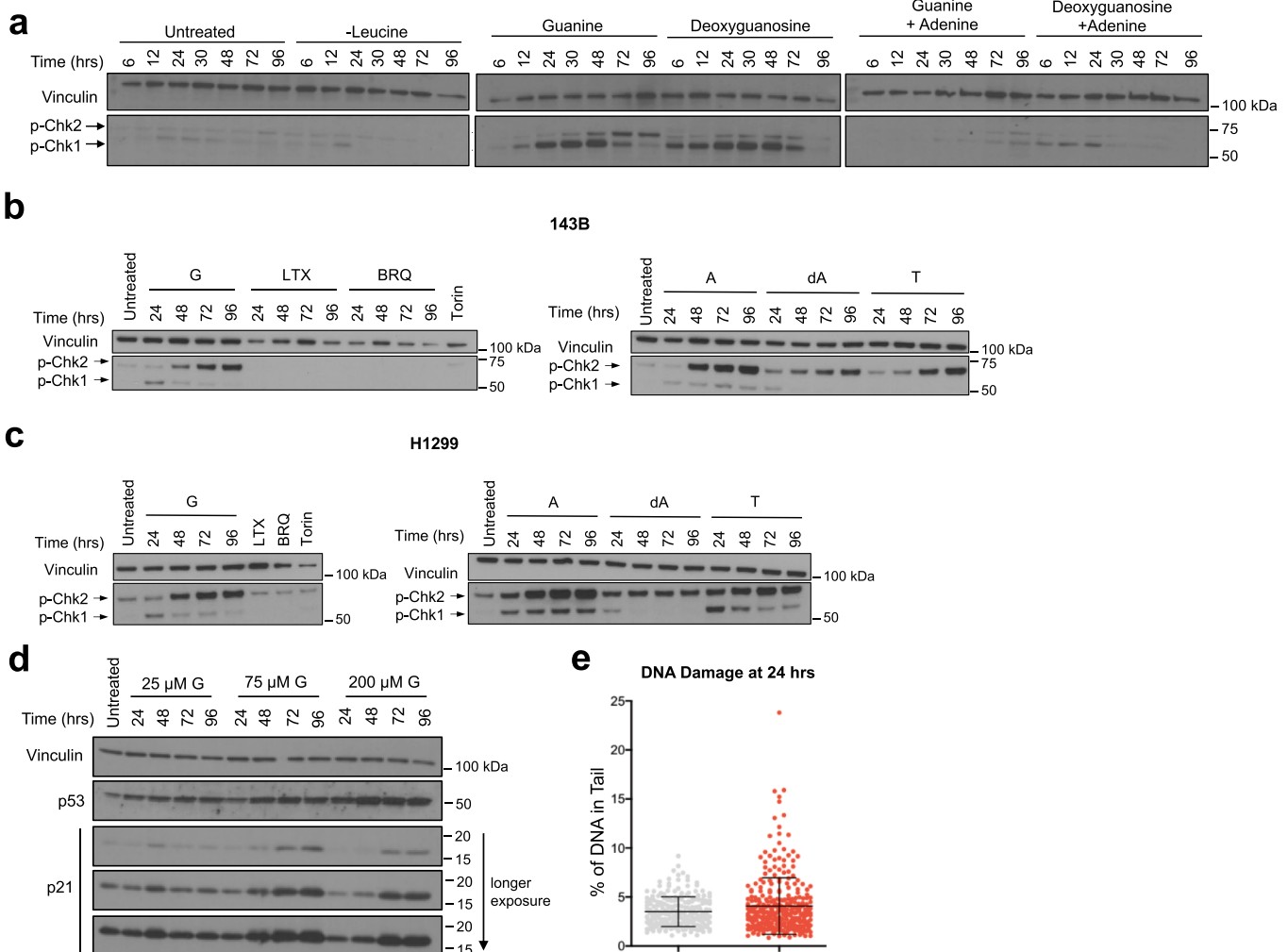

**Extended Data Fig. 5 | Imbalanced nucleotides induce replication stress signaling. a**, Western blot assessing phosphorylation of Chk1 and Chk2 in A549 cells cultured for the indicated time in standard media (Untreated) or media lacking leucine, or with addition of 200 μM guanine or 20 μM deoxyguanosine, with or without addition of 200 μM adenine as indicated. Levels of vinculin are also shown as a loading control. **b**, Western blot assessing phosphorylation of Chk1 and Chk2 in 143B cells cultured in standard conditions (Untreated) or treated with 200 μM guanine (G), 1 μM lometrexol (LTX), 1 μM brequinar (BRQ), 1 mM thymidine (T), 2.5 mM adenine (A), or 1.5 mM deoxyadenosine (dA) for the indicated time, or treated with 1 μM Torin for 24 hours. Levels of vinculin are also shown as a loading control. **c**, Western blot assessing phosphorylation of Chk1 and Chk2 in H1299 cells cultured in standard conditions (Untreated), treated with 200 μM G, 1 mM T, 2.5 mM A, or 1.5 mM dA for the indicated time, treated with 1 μM Torin for 24 hours, or treated with 1 μM LTX or 1 μM BRQ for 96 hours. Levels of vinculin are also shown as a loading control. **d**, Western blot showing levels of p53 and p21 in A549 cells in standard media (Untreated) or cultured for the indicated time with the indicated concentration of G. Levels of vinculin are also shown as a loading control. **e**, Comet assay to assess the presence of both single-stranded DNA and double-stranded DNA breaks (DNA damage) in A549 cells treated without (Untreated) or with 200 μM guanine for 24 hours. 529 cells were analyzed. Data are presented as mean +/- SD. Numerical source data and unprocessed blots are available in source data.

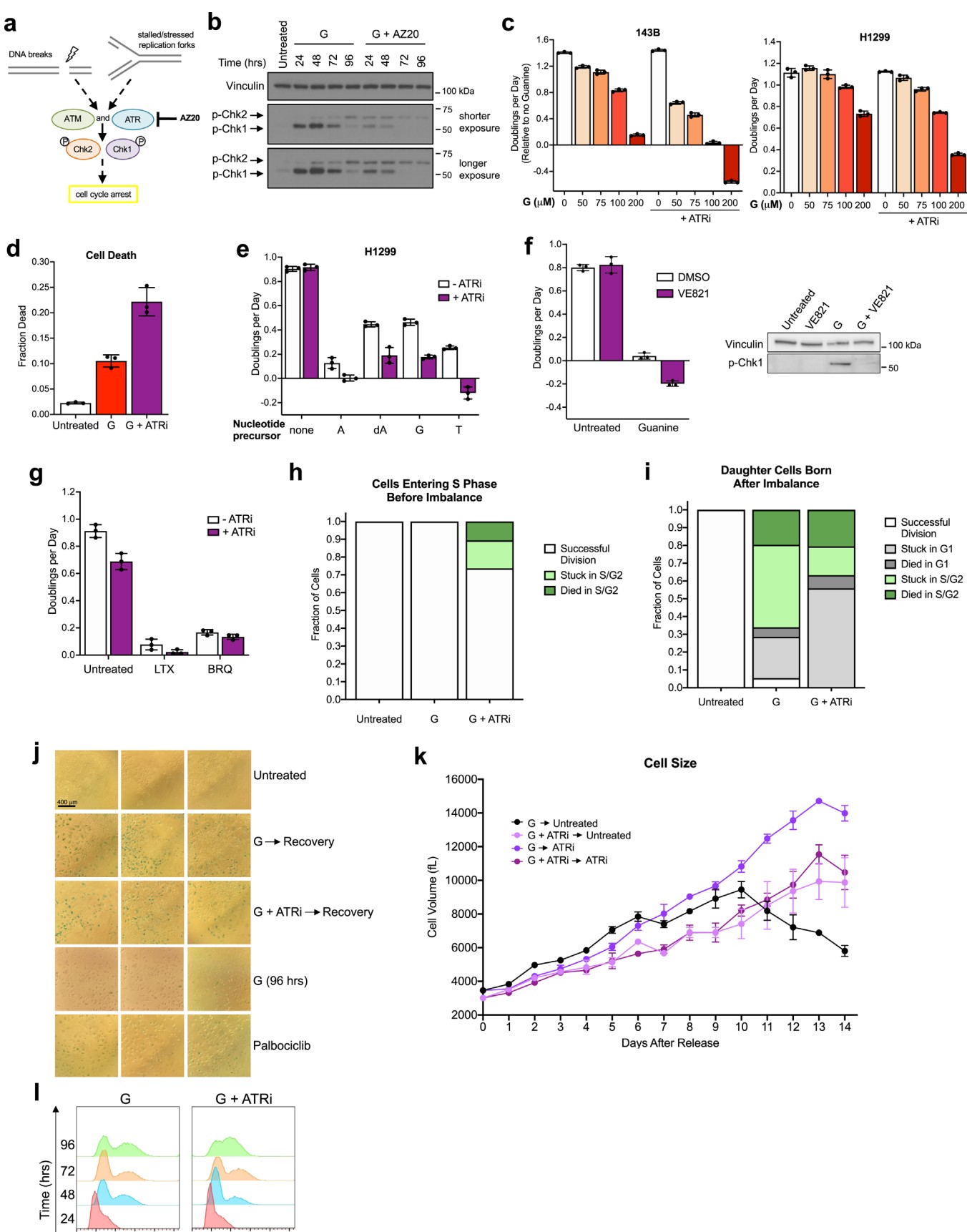

**Extended Data Fig. 6 | See next page for caption.**

**Extended Data Fig. 6 | ATR signaling impacts the fate of cells with imbalanced nucleotides. a**, Schematic outlining how ATR and ATM kinases respond to replication stress and DNA damage. The ATR and ATM targets Chk1 and Chk2 activate downstream effectors that halt cell cycle progression. AZ20 is an inhibitor of ATR kinase activity. **b**, Western blot assessing phosphorylation of Chk1 and Chk2 in A549 cells cultured in standard media (Untreated) or treated for the indicated time with 200 μM guanine (G) with or without 50 nM AZ20. Levels of vinculin are also shown as a control. **c**, Proliferation rates of 143B and H1299 cells treated with the indicated concentration of G with or without 50 nM AZ20 (ATRi). **d**, Cell death measured in A549 cells cultured for 96 hours in standard conditions (Untreated) or treated with 200 μM G with or without 50 nM ATRi as indicated. **e**, Proliferation rates of H1299 cells cultured in standard conditions (none) or treated with 2 mM adenine (A), 1.5 mM deoxyadenosine (dA), 200 μM G, or 1 mM thymidine (T), with or without 50 nM ATRi as indicated. **f**. Proliferation rates (left) and Western blot assessing phosphorylation of Chk1 (right) in A549 cells treated with or without 200 μM G with or without 0.6 μM of the ATR inhibitor VE821. Levels of vinculin are also shown as a control. **g**, Proliferation rates of A549 cells cultured in standard conditions (Untreated) or treated with 1 μM lometrexol (LTX) or 1 μM brequinar (BRQ) with or without 50 nM AZ20 (ATRi) as indicated. **h**, Cell fate as assessed using live-cell imaging of A549 mother cells expressing the mVenus-Gem1 reporter that were in S/G2 phase at the time of addition of 200 μM G with or without 50 nM AZ20 (ATRi). The fate of mother cells in S/G2 not exposed to excess G is also shown (Untreated). 83 cells were analyzed. **i**, Cell fate as assessed using live-cell imaging of A549 daughter cells expressing the mVenus-Gem1 reporter that were born after the addition of 200 μM G with or without 50 nM AZ20 (ATRi). The fate of daughter cells not exposed to excess G is also shown (Untreated). 158 cells were analyzed. **j**. SA-β-galactosidase activity was assayed in A549 cells cultured in standard conditions (Untreated), treated with 200 μM G for 96 hrs, or treated with 200 μM G with or without 50 nM AZ20 (ATRi) for 96 hrs and then switched to untreated media for 7 days (Recovery). Cells treated with Palbociclib for 7 days were included as a control. **k**, Mean volume of A549 cells measured over time after release from treatment with G with or without AZ20 (ATRi) as described in Fig. 5h. **l**, Cell cycle distribution as assessed by DNA content of A549 cells treated with 200 μM G with or without 50 nM AZ20 (ATRi) for the indicated time. Data are presented as mean ± SD of 3 biological replicates. Source numerical data and unprocessed blots are available in source data.

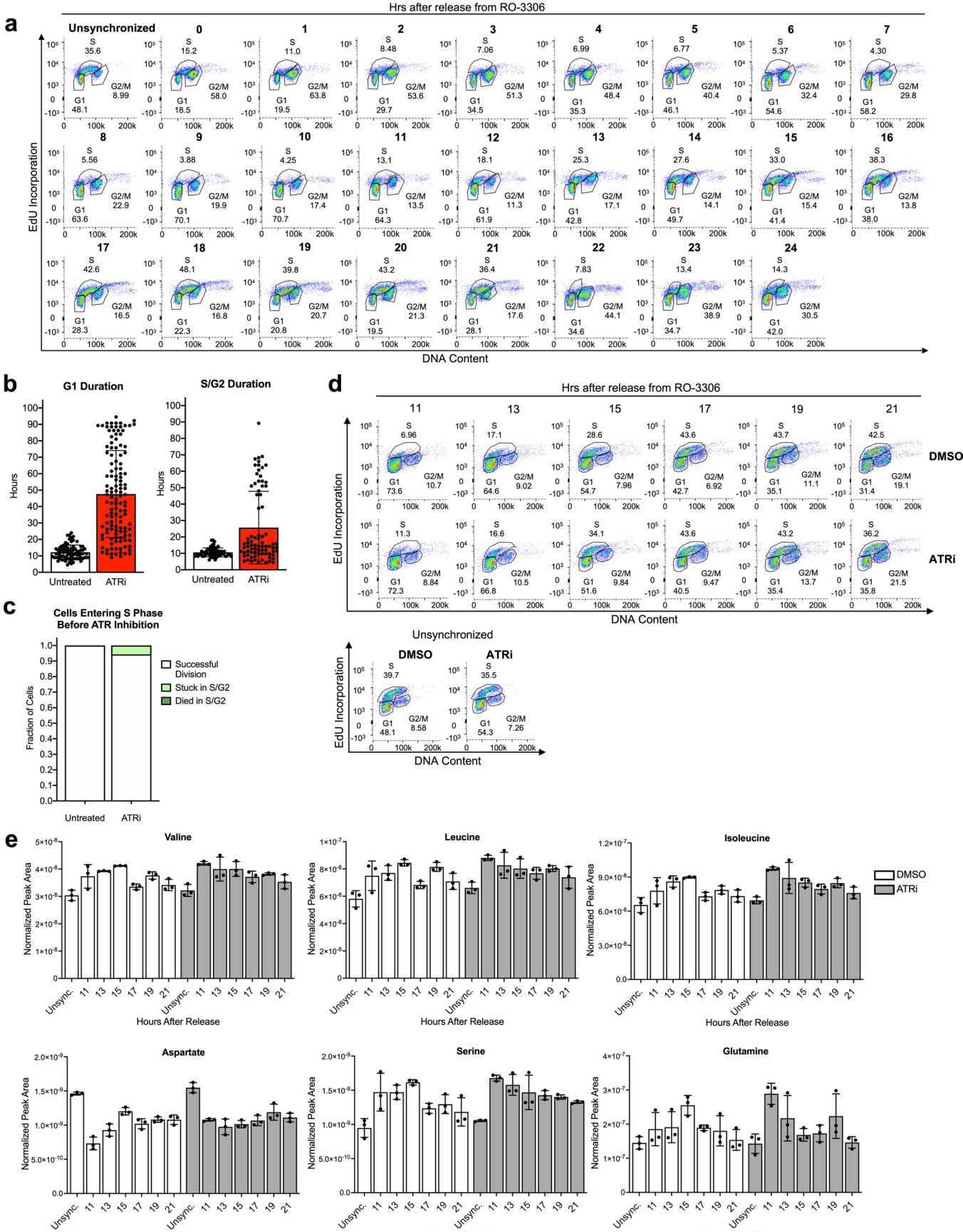

**Extended Data Fig. 7 | See next page for caption.**

**Extended Data Fig. 7 | Amino acid levels remain relatively constant during unperturbed cell cycles and are not impacted by loss of ATR signaling.**
**a**, Cell cycle distribution of A549 cells corresponding to Western blots shown in Fig. 6b. Cells were arrested in G2 phase by treating with 9 μM RO-3306 for 18 hours, then RO-3306 was removed to release from cell cycle arrest for the indicated time. A549 cells in standard culture are also shown (Unsynchronized). Cells were pulsed with EdU for 30 minutes prior to each time point and analyzed as outlined in Fig. 3a. **b**, Duration of G1 phase and S/G2 phases in A549 cells expressing the mVenus-Gem1 reporter cultured in standard conditions (Untreated) or with 50 nM AZ20 (ATRi), as assessed using live-cell imaging. 87 cells were analyzed. **c**, Cell fate as assessed using live-cell imaging of A549 mother cells expressing the mVenus-Gem1 reporter that were in S/G2 phase when 50 nM ATRi was added. The fate of mother cells in S/G2 not exposed to ATRi is also shown (Untreated). 55 cells were analyzed. **d**, Cell cycle distribution of A549 cells corresponding to the metabolite measurements shown in panel e, and in Fig. 6e and 6f. Cells were arrested in G2 phase by treating with 4.5 μM RO-3306 for 18 hours, then RO-3306 was removed to release from cell cycle arrest for the indicated time. Cells were pulsed with EdU for 30 minutes prior to each time point and analyzed as outlined in Fig. 3a. Unsynchronized cells were treated with DMSO or 50 nM ATRi for 24 hours as indicated. **e**, Levels of the indicated amino acids in A549 cells synchronized in G2 phase by treating with 4.5 μM RO-3306 for 18 hours, then released into the cell cycle for the indicated time. At the time of release from RO-3306, cells were either treated with DMSO or 50 nM ATRi as indicated. Unsynchronized cells (Unsync.) were treated with DMSO or 50 nM ATRi for 24 hours as indicated. All metabolite levels were measured by LCMS. Data are presented as mean ± SD of 3 biological replicates. Source numerical data are available in source data.

# Reporting Summary

## Statistics

For all statistical analyses, confirm that the following items are present in the figure legend, table legend, main text, or Methods section.

| n/a | Confirmed | |
|---|---|---|
| ☐ | ☒ | The exact sample size (*n*) for each experimental group/condition, given as a discrete number and unit of measurement |
| ☐ | ☒ | A statement on whether measurements were taken from distinct samples or whether the same sample was measured repeatedly |
| ☒ | ☐ | The statistical test(s) used AND whether they are one- or two-sided<br>*Only common tests should be described solely by name; describe more complex techniques in the Methods section.* |
| ☒ | ☐ | A description of all covariates tested |
| ☒ | ☐ | A description of any assumptions or corrections, such as tests of normality and adjustment for multiple comparisons |
| ☐ | ☒ | A full description of the statistical parameters including central tendency (e.g. means) or other basic estimates (e.g. regression coefficient) AND variation (e.g. standard deviation) or associated estimates of uncertainty (e.g. confidence intervals) |
| ☒ | ☐ | For null hypothesis testing, the test statistic (e.g. *F*, *t*, *r*) with confidence intervals, effect sizes, degrees of freedom and *P* value noted<br>*Give P values as exact values whenever suitable.* |
| ☒ | ☐ | For Bayesian analysis, information on the choice of priors and Markov chain Monte Carlo settings |
| ☒ | ☐ | For hierarchical and complex designs, identification of the appropriate level for tests and full reporting of outcomes |
| ☒ | ☐ | Estimates of effect sizes (e.g. Cohen's *d*, Pearson's *r*), indicating how they were calculated |

*Our web collection on statistics for biologists contains articles on many of the points above.*

## Software and code

Policy information about availability of computer code

| | |
|---|---|
| Data collection | BD FACSDiva v9.0 software was used to collect flow cytometry data. Live cell imaging data was collected using IncuCyte S3 2017A live cell imaging system (Sartorius). |
| Data analysis | Flow cytometry data was analyzed using FlowJo v10 software. Mass spectrometry data was analyzed with XCalibur QuanBrowser 2.2. Comet assay data was anlyzed with ImageJ with OpenComet v1.3 software. |

For manuscripts utilizing custom algorithms or software that are central to the research but not yet described in published literature, software must be made available to editors and reviewers. We strongly encourage code deposition in a community repository (e.g. GitHub). See the Nature Portfolio guidelines for submitting code & software for further information.

## Data

Policy information about availability of data

All manuscripts must include a data availability statement. This statement should provide the following information, where applicable:
- Accession codes, unique identifiers, or web links for publicly available datasets
- A description of any restrictions on data availability
- For clinical datasets or third party data, please ensure that the statement adheres to our policy

Numerical source data and unprocessed blots are available in the source data files for their corresponding figures.

# Field-specific reporting

Please select the one below that is the best fit for your research. If you are not sure, read the appropriate sections before making your selection.

☒ Life sciences          ☐ Behavioural & social sciences          ☐ Ecological, evolutionary & environmental sciences

For a reference copy of the document with all sections, see nature.com/documents/nr-reporting-summary-flat.pdf

# Life sciences study design

All studies must disclose on these points even when the disclosure is negative.

| | |
|---|---|
| Sample size | Experiments were performed using standard sample sizes that produced tolerable error with respect to effect size. No statistical test was performed to predetermine sample size. |
| Data exclusions | No data was excluded. |
| Replication | Experiments were repeated in biological triplicates. |
| Randomization | This is not relevant to the study, as all experiments were done using human cell lines. No experiments involved allocation of different samples, organisms, or participants into experimental groups. |
| Blinding | Blinding was not relevant to the study because no experiments involved allocation of different samples, organisms, or participants into experimental groups. |

# Reporting for specific materials, systems and methods

We require information from authors about some types of materials, experimental systems and methods used in many studies. Here, indicate whether each material, system or method listed is relevant to your study. If you are not sure if a list item applies to your research, read the appropriate section before selecting a response.

## Materials & experimental systems

| n/a | Involved in the study |
|---|---|
| ☐ | ☒ Antibodies |
| ☐ | ☒ Eukaryotic cell lines |
| ☒ | ☐ Palaeontology and archaeology |
| ☒ | ☐ Animals and other organisms |
| ☒ | ☐ Human research participants |
| ☒ | ☐ Clinical data |
| ☒ | ☐ Dual use research of concern |

## Methods

| n/a | Involved in the study |
|---|---|
| ☒ | ☐ ChIP-seq |
| ☐ | ☒ Flow cytometry |
| ☒ | ☐ MRI-based neuroimaging |

## Antibodies

| | |
|---|---|
| Antibodies used | Primary antibodies: vinculin (Cell Signaling Technology #4650), phospho-ribosomal protein S6 Ser 235/236 (CST #4858), ribosomal protein S6 (CST #2217), phospho-p70 S6 kinase Thr389 (CST #9205), p70 S6 kinase (CST #9202), phospho-Akt (CST #4060), Akt (CST #9272), phospho-AMPK (CST #2535), AMPK (CST #2532), phospho-Chk1 Ser345 (CST #2341), phospho-Chk1 Ser345 (CST #2348), phospho-Chk2 Thr68 (CST,2197), p53 (CST #9282), and p21 (CST, #2947). Primary antibodies were used at a dilution of 1:1000 Secondary antibodies: Anti-rabbit (CST #7074), anti-mouse (CST #7076). Anti-rabbit was used at a 1:5000 dilution and anti-mouse was used at a dilution of 1:10000. |
| Validation | Antibodies were validated by the suppliers and by the appearance of a band at the predicted size. The manufacturers provide the following statement on their multiple approaches to antibody validation: "To ensure our antibodies will work in your experiment, we adhere to the Hallmarks of Antibody Validation™, six complementary strategies that can be used to determine the functionality, specificity, and sensitivity of an antibody in any given assay. CST adapted the work by Uhlen, et. al., ("A Proposal for Validation of Antibodies." Nature Methods (2016)) to build the Hallmarks of Antibody Validation, based on our decades of experience as an antibody manufacturer and our dedication to reproducible science." Antibodies for phospho-ribosomal protein S6 Ser 235/236 and phospho-p70 S6 kinase Thr389 were also validated by treating cells with the mTORC1 inhibitor Torin1 and observing a decrease in signal. |

## Eukaryotic cell lines

Policy information about cell lines

| | |
|---|---|
| Cell line source(s) | Cell lines were obtained from ATCC (catalog numbers: A549 – CCL-185; H1299 – CRL-5803; 143B – CRL-8303; U2OS – HTB-96; |

| Cell line source(s) | MDA-MB-468 – HTB-132; RPE-1 – CRL-4000); A9 cells were a gift from the B. Manning laboratory (original source was ATCC, catalog number CCL-1.4) |
|---|---|
| Authentication | The identities of all cells were authenticated by satellite tandem repeat testing and referenced to ATCC values. |
| Mycoplasma contamination | All cell lines tested negatively for mycoplasma. |
| Commonly misidentified lines (See ICLAC register) | None of the cell lines used in this study are listed as commonly misidentified cell lines. |

# Flow Cytometry

## Plots

Confirm that:

☒ The axis labels state the marker and fluorochrome used (e.g. CD4-FITC).

☒ The axis scales are clearly visible. Include numbers along axes only for bottom left plot of group (a 'group' is an analysis of identical markers).

☒ All plots are contour plots with outliers or pseudocolor plots.

☒ A numerical value for number of cells or percentage (with statistics) is provided.

## Methodology

| Sample preparation | Samples were prepared from cultured human cell lines. To fix cells for staining and flow cytometry, cells were trypsinized, pelleted and washed twice with PBS. Cells were resuspended in 500 µL ice-cold PBS and 5 mL ice-cold ethanol was added dropwise to each sample while vortexing in order to obtain a single-cell suspension. |
|---|---|
| Instrument | A BD FACSCanto II Cell Analyzer was used to collect data. |
| Software | BD FACSDiva Software was used to collect the data, and FlowJo Software was used to analyze the data. |
| Cell population abundance | This is not relevant because cells were not sorted. |
| Gating strategy | Cells were gated based on FSC and SSC to distinguish live cells for analysis. |

☒ Tick this box to confirm that a figure exemplifying the gating strategy is provided in the Supplementary Information.

