## [Peer Review File · Nature Cell Biology]

Peer Review Information

Journal: Nature Cell Biology

Manuscript Title: Nucleotide imbalance decouples cell growth from cell proliferation

Corresponding author name(s): Matthew Vander Heiden

Reviewer Comments & Decisions:

Decision Letter, initial version:
--

Dear Dr Vander Heiden,

I'm writing on behalf of my colleague Melina Casadio, who is currently out of the office.

Your manuscript, "Nucleotide imbalance decouples cell growth from cell proliferation", has now been seen by 3 referees, who are experts in nucleotide metabolism (referee 1); cell cycle and replication stress (referee 2); and replication stress (referee 3). As you will see from their comments (attached below) they find this work of potential interest, but have raised substantial concerns, which in our view would need to be addressed with considerable revisions before we can consider publication in Nature Cell Biology.

Nature Cell Biology editors discuss the referee reports in detail within the editorial team, including the chief editor, to identify key referee points that should be addressed with priority, and requests that are overruled as being beyond the scope of the current study. To guide the scope of the revisions, I have listed these points below. I should stress that the referees' concerns point to a premature dataset and these points would need to be addressed with experiments and data, and reconsideration of the study for this journal and re-engagement of referees would depend on strength of these revisions.

In particular, it would be essential to:

- a) define nucleotide imbalance in a specific and quantitative way and investigate whether nucleotide imbalance or dNTP reduction is the cause of replication stress and ATR activation, as noted by referee 3 (points 1, 4, 5, 8).
- b) address the concern on the use of a single cell line raised by referee 2.
- c) All other referee concerns pertaining to strengthening existing data, providing controls, methodological details, clarifications and textual changes should also be addressed.

d) Finally please pay close attention to our guidelines on statistical and methodological reporting (listed below) as failure to do so may delay the reconsideration of the revised manuscript. In particular please provide:

We would be happy to consider a revised manuscript that would satisfactorily address these points, unless a similar paper is published elsewhere, or is accepted for publication in Nature Cell Biology in the meantime.

- ensure that it conforms to our format instructions and publication policies (see below and <https://www.nature.com/nature/for-authors>).

- provide a point-by-point rebuttal to the full referee reports verbatim, as provided at the end of this letter.

- provide the completed Reporting Summary (found here <https://www.nature.com/documents/nr-reporting-summary.pdf>). This is essential for reconsideration of the manuscript will be available to editors and referees in the event of peer review. For more information see <http://www.nature.com/authors/policies/availability.html> or contact me.

When submitting the revised version of your manuscript, please pay close attention to our [href="https://www.nature.com/nature-research/editorial-policies/image-integrity">Digital Image Integrity Guidelines](https://www.nature.com/nature-research/editorial-policies/image-integrity). and to the following points below:

Finally, please ensure that you retain unprocessed data and metadata files after publication, ideally archiving data in perpetuity, as these may be requested during the peer review and production process

2or after publication if any issues arise.

Nature Cell Biology is committed to improving transparency in authorship. As part of our efforts in this direction, we are now requesting that all authors identified as 'corresponding author' on published papers create and link their Open Researcher and Contributor Identifier (ORCID) with their account on the Manuscript Tracking System (MTS), prior to acceptance. ORCID helps the scientific community achieve unambiguous attribution of all scholarly contributions. You can create and link your ORCID from the home page of the MTS by clicking on 'Modify my Springer Nature account'. For more information please visit www.springernature.com/orcid.

This journal strongly supports public availability of data. Please place the data used in your paper into a public data repository, or alternatively, present the data as Supplementary Information. If data can only be shared on request, please explain why in your Data Availability Statement, and also in the correspondence with your editor. Please note that for some data types, deposition in a public repository is mandatory - more information on our data deposition policies and available repositories appears below.

[REDACTED]

We would like to receive a revised submission within six months.

We hope that you will find our referees' comments, and editorial guidance helpful. Please do not hesitate to contact me if there is anything you would like to discuss.

Best wishes,

Jie Wang

Jie Wang, PhD
Senior Editor
Nature Cell Biology

Tel: +44 (0) 207 843 4924
email: jie.wang@nature.comReviewers' Comments:

Reviewer #1:

Remarks to the Author:

This is an elegant study that thoroughly examines the impact of nucleotide imbalance on cellular and metabolic fitness. The authors found that supraphysiological supplementation of single nucleotide precursors (e.g., guanine, adenine) lead to a general purine nucleotide imbalance accompanied by cell cycle arrest and a halt in cell proliferation. Interestingly, they found that the activity of major growth factor signaling pathways, such as mTORC1 and Akt, were not altered, which allowed these cells to grow despite being unable to undergo cell division, thus uncoupling cell growth from cell proliferation. Instead, the authors identify that nucleotide imbalance triggers the ATR replication stress-sensing which supports dNTP availability and allows for cell survival. The study is of high quality, rigorous, and clearly written.

A few minor comments can help to improve the presentation of the study.

- 1) The authors observe a robust cell cycle arrest in response to nucleotide imbalance. Is there a distinct effect between purine versus pyrimidine imbalance concerning cell cycle markers?
- 2) The authors propose that nucleotide imbalance might cause a senescence phenotype. Do the authors observe any effect on senescence markers (e.g., beta-galactosidase, p16, gamma-H2AX) in response to purine or pyrimidine imbalance?
- 3) It would be beneficial for the reader to have quantification of the FACS scatter plots.

Reviewer #2:

Remarks to the Author:

"Nucleotide imbalance decouples cell growth from cell proliferation"

The authors present experiments that address the impact of nucleosides added to the tissue culture media on the growth and proliferation of a human cancer cell line (A549) in vitro. This is original and significant.

The authors have data that suggests that nucleosides impact S phase and the cell cycle rather than growth. The initial experiments are interesting (Fig 1) and definitely worth pursuing, but, in my view, the experiments designed to address the observations in Figure 1 are difficult to interpret as insufficient technical detail is provided. I cannot describe the data as robust, valid, or reliable as technical details, particularly for the LC-MS/MS are not provided.

Ultimately the experiments fall short of mechanism, are limited by the use of a single cancer cell line, and the interpretation of some of the experiments is difficult as technical details are either not provided or are not performed. The experiments fall a long way short of the separating the impact of exogenous nucleosides on RNA transcription and DNA replication.

Below I list what I believe are strengths and weaknesses in the manuscript:

Throughout the manuscript nucleoside and nucleotide are confused.

4Figure 1 - the experiment should be presented precisely, without assumptions. The data are interesting. Addition of nucleoside to the media impacts proliferation. You have no data as to whether that nucleoside impacts the concentration of nucleotides in the cell in Figure 1.

Figure 2 - no standard curves are provided for the LC-MS/MS (?) used to quantitate intracellular nucleotide concentrations in cells. This is arguably the most important component of the manuscript and I find it wanting. What is the impact of nucleosides in the media on intracellular nucleotide concentrations in cells (and then does this impact transcription or replication). The authors state themselves that they cannot distinguish dGTP from ATP. No documentation about resolution and quantitation of nucleotides is provided. It is generally accepted that this is challenging. Details must be provided.

Figure 3 - it is reasonable to hypothesize that 200 μ M G may impact the transport of EdU and subsequent intracellular concentration of EdU in the cell. This is ignored. There is a similar concern in Figure 6 where an ATRi is used (a competitive ATP inhibitor - an ATP analogue).

Figure 4 is excellent. Nucleosides in the media do not appear to impact translation and growth. It would be helpful to look at transcription.

Figure 5 is excellent. Nucleosides in the media do impact DNA replication insofar as they induce DNA damage signaling that is generally (canonical signaling always) initiated at replication forks.

Figure 7 is excellent.

In summary, there are clear strengths in the paper and the ideas presented are provocative and interesting. However, the strengths are outweighed by the technical weaknesses, the absence of mechanism, and the use of a single cancer cell line. Finally, the manuscript is not lucid - it is actually quite hard to follow.

Reviewer #3:

Remarks to the Author:

In this manuscript, the authors show that several precursors of nucleotides can inhibit cell proliferation in a manner dependent on the salvage pathway. This observation is consistent with the known effects of thymidine. Furthermore, the authors show that guanine increases the synthesis of GTP through the salvage pathway but inhibits the de novo synthesis of both GTP and ATP, leading to an increase in GTP but a reduction in ATP and an imbalance between GTP and ATP. Adding adenine to cells restored ATP levels and reversed the GTP/ATP imbalance in guanine treated cells, and adenine also overcame the inhibition of cell proliferation by guanine.

The authors then carefully characterized how guanine affects cell proliferation during the cell cycle. They show that guanine clearly interferes with DNA replication in S phase and causes a delay in S/G2. Unexpectedly, however, guanine does not inhibit mTOR and protein synthesis, and cell growth is

5uncoupled from cell proliferation in cells with nucleotide imbalance. These results suggest that cells do not rely on the mTOR pathway to sense nucleotide imbalance.

Finally, the authors show that guanine treatment activates the ATR/ATM pathway in S phase, and that inhibition of ATR leads to irreversible DNA damage in cells with imbalanced nucleotides. They also provide evidence that ATR is activated during the unperturbed cell cycle, and that ATR promotes dNTP synthesis during S phase. Based on these results, the authors suggest that ATR plays a key role in sensing nucleotide imbalance and prevents the uncoupling of cell growth and cell proliferation.

This manuscript contains many interesting observations, and the model proposed is quite attractive. However, 'nucleotide imbalance' is not defined in a specific and quantitative way. Whether nucleotide imbalance or reduction in dNTP is the cause of replication problems and ATR activation is not clearly addressed. The known effects of thymidine and known functions of ATR in DNA replication also reduce the novelty of the model. Additional experiments are needed to strengthen this study and make it suitable for NCB.

Specific comments.

1. How 'nucleotide imbalance' is defined is not clear. As shown in Fig. 2d, many changes of nucleotides are observed after cells are treated with nucleotide precursors. What exactly is a 'nucleotide imbalance'? In the case of guanine, it is clear that the GTP/ATP balance is altered (Fig. 2a). However, in the case of thymidine, neither GTP/ATP nor UTP/CTP ratios change much, but cell proliferation is clearly inhibited (Fig. 2f). It is unclear to me whether 'nucleotide imbalance' refers to the imbalance between specific nucleotides or the overall imbalance among multiple nucleotides. Having a clear and quantitative definition of nucleotide imbalance is important for the model.
2. In Fig. 2c and extended Fig. 2, A inhibited the salvage pathway for G, but G did not inhibit the salvage pathway for A. How can this difference between A and G be explained?
3. Fig. 2a and 2e suggest that A reverses the effects of G on cell proliferation. Can one expect that G also reverses the effects of A on cell proliferation? Can the authors test whether G reverses the nucleotide changes caused by A?
4. In Fig. 2f, the rescue of T treated cells by C is not explained. Does C change any nucleotides affected by T? What is the key nucleotide imbalance that inhibits the proliferation of T treated cells? Thymidine is known to reduce dCTP. Is rescuing effect of C simply attributed to an increase of dCTP?
5. In Fig. 2g, it is clearly that guanine caused severe reductions in dTTP and dCTP. These changes are also observed in S phase after guanine treatment (Fig. 3k). Are these changes in dTTP and dCTP dependent on the salvage of G? Can these effects of G be reversed by A? If guanine reduces dTTP and dCTP, it becomes difficult to tell whether the S phase problems caused by guanine are attributed to nucleotide imbalance or the reduction in dNTPs. This is a conceptually important question to the model. If dNTP reduction is the cause of replication inhibition, the effects of guanine would become quite similar to those of thymidine, which is known to inhibit replication by reducing dCTP.

6. In Fig. 3j and 3k, one could argue that dTTP and dCTP are reduced because G treated cells did not progress through S phase efficiently. This appears to be a “chicken and egg” problem.

7. In Fig. 5d, it is surprising that LTX and BRQ did not activate ATR efficiently. LTX and BRQ can reduce DNA synthesis as efficiently as guanine (extend Fig. 3c). If DNA synthesis is severely compromised, why isn't ATR activated?

8. The data in Fig. 6 and 7 are consistent with the model in which ATR is important in cells with imbalanced nucleotides. However, dNTP levels are also changed in these cells. It is unclear whether nucleotide imbalance or reduction in dNTP is the cause of ATR dependency.

9. The novelty of Fig. 7 may be limited. Recent studies have shown that ATR plays an important role in unperturbed early S phase to limit replication origin firing and promote RRM2 accumulation (Buisson et al. Mol Cell 2015). The observations in Fig. 7 are quite consistent with the previous model and provide additional evidence on the changes of dNTPs in S phase.

10. Thymidine is known to activate the salvage pathway and indirectly affect dCTP levels. Conceptually, this is quite similar to what is proposed in this study. Perhaps it is not surprising that nucleotide imbalance would affect dNTP levels and indirectly cause replication stress and inhibition of cell proliferation. The novel finding of this study is that nucleotide imbalance is not detected by the mTOR pathway, unlike the depletion of nucleotides. It would be important to better explain how even imbalanced nucleotides can activate mTOR and how mTOR promotes S phase entry in this context.

MANUSCRIPT FORMAT – please follow the guidelines listed in our Guide to Authors regarding manuscript formats at Nature Cell Biology.TITLE – should be no more than 100 characters including spaces, without punctuation and avoiding technical terms, abbreviations, and active verbs..

Methods should be written concisely, but should contain all elements necessary to allow interpretation

8and replication of the results. As a guideline, Methods sections typically do not exceed 3,000 words. The Methods should be divided into subsections listing reagents and techniques. When citing previous methods, accurate references should be provided and any alterations should be noted. Information must be provided about: antibody dilutions, company names, catalogue numbers and clone numbers for monoclonal antibodies; sequences of RNAi and cDNA probes/primers or company names and catalogue numbers if reagents are commercial; cell line names, sources and information on cell line identity and authentication. Animal studies and experiments involving human subjects must be reported in detail, identifying the committees approving the protocols. For studies involving human subjects/samples, a statement must be included confirming that informed consent was obtained. Statistical analyses and information on the reproducibility of experimental results should be provided in a section titled "Statistics and Reproducibility".

All Nature Cell Biology manuscripts submitted on or after March 21 2016 must include a Data availability statement as a separate section after Methods but before references, under the heading "Data Availability". For Springer Nature policies on data availability see <http://www.nature.com/authors/policies/availability.html>; for more information on this particular policy see <http://www.nature.com/authors/policies/data/data-availability-statements-data-citations.pdf>. The Data availability statement should include:

- Accession codes for primary datasets (generated during the study under consideration and designated as "primary accessions") and secondary datasets (published datasets reanalysed during the study under consideration, designated as "referenced accessions"). For primary accessions data should be made public to coincide with publication of the manuscript. A list of data types for which submission to community-endorsed public repositories is mandated (including sequence, structure, microarray, deep sequencing data) can be found here <http://www.nature.com/authors/policies/availability.html#data>.
- Unique identifiers (accession codes, DOIs or other unique persistent identifier) and hyperlinks for datasets deposited in an approved repository, but for which data deposition is not mandated (see here for details <http://www.nature.com/sdata/data-policies/repositories>).
- At a minimum, please include a statement confirming that all relevant data are available from the authors, and/or are included with the manuscript (e.g. as source data or supplementary information), listing which data are included (e.g. by figure panels and data types) and mentioning any restrictions on availability.
- If a dataset has a Digital Object Identifier (DOI) as its unique identifier, we strongly encourage including this in the Reference list and citing the dataset in the Methods.

We recommend that you upload the step-by-step protocols used in this manuscript to the Protocol Exchange. More details can found at www.nature.com/protocolexchange/about.

DISPLAY ITEMS – main display items are limited to 6-8 main figures and/or main tables for Articles, Resources, Technical Reports; and 5 main figures and/or main tables for Letters. For Supplementary Information see below.FIGURES – Colour figure publication costs \$600 for the first, and \$300 for each subsequent colour figure. All panels of a multi-panel figure must be logically connected and arranged as they would appear in the final version. Unnecessary figures and figure panels should be avoided (e.g. data presented in small tables could be stated briefly in the text instead).

All imaging data should be accompanied by scale bars, which should be defined in the legend. Cropped images of gels/blots are acceptable, but need to be accompanied by size markers, and to retain visible background signal within the linear range (i.e. should not be saturated). The boundaries of panels with low background have to be demarked with black lines. Splicing of panels should only be considered if unavoidable, and must be clearly marked on the figure, and noted in the legend with a statement on whether the samples were obtained and processed simultaneously. Quantitative comparisons between samples on different gels/blots are discouraged; if this is unavoidable, it should only be performed for samples derived from the same experiment with gels/blots were processed in parallel, which needs to be stated in the legend.

- For line art, graphs, charts and schematics we prefer Adobe Illustrator (.AI), Encapsulated PostScript (.EPS) or Portable Document Format (.PDF). Files should be saved or exported as such directly from the application in which they were made, to allow us to restyle them according to our journal house style.
- We accept PowerPoint (.PPT) files if they are fully editable. However, please refrain from adding PowerPoint graphical effects to objects, as this results in them outputting poor quality raster art. Text used for PowerPoint figures should be Helvetica (preferred) or Arial.
- We do not recommend using Adobe Photoshop for designing figures, but we can accept Photoshop generated (.PSD or .TIFF) files only if each element included in the figure (text, labels, pictures, graphs, arrows and scale bars) are on separate layers. All text should be editable in 'type layers' and line-art such as graphs and other simple schematics should be preserved and embedded within 'vector smart objects' - not flattened raster/bitmap graphics.
- Some programs can generate Postscript by 'printing to file' (found in the Print dialogue). If using an application not listed above, save the file in PostScript format or email our Art Editor, Allen Beattie for advice (a.beattie@nature.com).

10Regardless of format, all figures must be vector graphic compatible files, not supplied in a flattened raster/bitmap graphics format, but should be fully editable, allowing us to highlight/copy/paste all text and move individual parts of the figures (i.e. arrows, lines, x and y axes, graphs, tick marks, scale bars etc.). The only parts of the figure that should be in pixel raster/bitmap format are photographic images or 3D rendered graphics/complex technical illustrations.

The total number of Supplementary Figures (not including the “unprocessed scans” Supplementary Figure) should not exceed the number of main display items (figures and/or tables (see our Guide to

Authors and March 2012 editorial <http://www.nature.com/ncb/authors/submit/index.html#suppinfo>; <http://www.nature.com/ncb/journal/v14/n3/index.html#ed>). No restrictions apply to Supplementary Tables or Videos, but we advise authors to be selective in including supplemental data.

GUIDELINES FOR EXPERIMENTAL AND STATISTICAL REPORTING

REPORTING REQUIREMENTS – We are trying to improve the quality of methods and statistics reporting in our papers. To that end, we are now asking authors to complete a reporting summary that collects information on experimental design and reagents. The Reporting Summary can be found here <https://www.nature.com/documents/nr-reporting-summary.pdf> If you would like to reference the guidance text as you complete the template, please access these flattened versions at <http://www.nature.com/authors/policies/availability.html>.

Author Rebuttal to Initial comments

Response to reviewers' comments (Reviewers' comments in **bold italic** and our response in plain text):

Reviewer #1:

Remarks to the Author:

This is an elegant study that thoroughly examines the impact of nucleotide imbalance on cellular and metabolic fitness. The authors found that supraphysiological supplementation of single nucleotide precursors (e.g., guanine, adenine) lead to a general purine nucleotide imbalance accompanied by cell cycle arrest and a halt in cell proliferation. Interestingly, they found that the activity of major growth factor signaling pathways, such as mTORC1 and Akt, were not altered, which allowed these cells to grow despite being unable to undergo cell division, thus uncoupling cell growth from cell proliferation. Instead, the authors identify that nucleotide imbalance triggers the ATR replication stress-sensing which supports dNTP availability and allows for cell survival. The study is of high quality, rigorous, and clearly written.

We thank the Reviewer for the positive feedback on the manuscript, for their thoughtful comments, and for their time spent evaluating the data.

A few minor comments can help to improve the presentation of the study.

1) The authors observe a robust cell cycle arrest in response to nucleotide imbalance. Is there a distinct effect between purine versus pyrimidine imbalance concerning cell cycle markers?

This is an interesting question that we have addressed by analyzing how the cell cycle is affected when cells are treated with G-, A-, T-, and C-nucleotide precursors. In these experiments we did not find a consistent difference in cell cycle arrest that is dependent on which excess nucleotide was provided (see Extended Data Fig. 3e in the revised manuscript). We suspect that the lack of major difference in cell cycle progression among different nucleotide imbalances is because the ultimate mechanism of S phase arrest is impaired replication fork progression due to imbalanced dNTP levels. Indeed, we used absolute quantification by LCMS to confirm that each nucleotide imbalance causes different imbalances in the levels of dNTPs (see Fig. 2e,g and Extended Data Fig. 2e,g in the revised manuscript). While minor differences in cell cycle distribution are observed among different cell lines after 24 hours of exposure to different nucleotides, we speculate that this

13is due to cell type-specific differences in nucleotide transport or salvage enzyme expression. Importantly, the response to either purine or pyrimidine imbalances leads to replication stress (see Fig. 5c,d and Extended Data Fig. 5b,c in the revised manuscript), supporting a model where each different imbalance converges on inhibition of DNA replication during S phase.

2) The authors propose that nucleotide imbalance might cause a senescence phenotype. Do the authors observe any effect on senescence markers (e.g., beta-galactosidase, p16, gamma-H2AX) in response to purine or pyrimidine imbalance?

We examined SA-beta-galactosidase staining in guanine (G)-treated cells, as well as cells treated with G that were then allowed to recover with or without ATR inhibition for 7 days. We chose this time point because we observed that cell volume continued to increase after nucleotide imbalance was relieved and that mean cell volume was highest at this time point after recovery from imbalance. Because increased cell size is a well-characterized phenotype of senescence, we reasoned that the population of cells at this time point might contain the highest proportion of any senescent cells caused by nucleotide imbalance.

Consistent with a known link between replication stress and senescence, SA-beta-galactosidase staining was increased in cells treated with G and allowed to recover, consistent with a degree of senescence induction in these cells. Interestingly, cells treated with G for 4 days did not exhibit increased beta-galactosidase activity, indicating that senescence induction is likely not an immediate response to nucleotide imbalance. Rather, persistent replication stress following nucleotide imbalance may result in senescence induction over longer periods of time in some cells. These data are presented in Extended Data Fig. 6j and discussed in the revised manuscript.

3) It would be beneficial for the reader to have quantification of the FACS scatter plots.

We agree with the Reviewer that quantification of FACS data can be helpful, and in the revised manuscript we now include quantification of relevant cells populations in several figures (see Fig. 3e,j and Extended Data Fig. 3f, 7a,d in the revised manuscript). Specifically, we opted to display gates for figures showing synchronized cells, as this information helped us to identify time points at which certain cell cycle states were enriched to enable LCMS-based metabolite measurements. We also display gates for experiments where EdU was pulsed and then washed out, as these experiments yield more distinct populations of EdU-positive cells. For other experiments however, we are concerned that drawing strict gates for cell cycle populations could be misleading. Staining for DNA content and EdU incorporation results in a spread of cell populations instead of distinct separate populations (for example, Fig. 3b), and it can be difficult to distinguish those cells in very early or very late S phase without drawing arbitrary gates. In these cases, we feel it is more rigorous to present the data showing the population spread only to avoid suggesting that we can

14

clearly distinguish distinct groups of cells by this method. However, if the Reviewer or Editor feel strongly, we are happy to include quantified gated populations for any additional FACS plots.

Reviewer #2:

Remarks to the Author:

"Nucleotide imbalance decouples cell growth from cell proliferation"

The authors present experiments that address the impact of nucleosides added to the tissue culture media on the growth and proliferation of a human cancer cell line (A549) in vitro. This is original and significant.

The authors have data that suggests that nucleosides impact S phase and the cell cycle rather than growth. The initial experiments are interesting (Fig 1) and definitely worth pursuing, but, in my view, the experiments designed to address the observations in Figure 1 are difficult to interpret as insufficient technical detail is provided. I cannot describe the data as robust, valid, or reliable as technical details, particularly for the LC-MS/MS are not provided.

Ultimately the experiments fall short of mechanism, are limited by the use of a single cancer cell line, and the interpretation of some of the experiments is difficult as technical details are either not provided or are not performed. The experiments fall a long way short of the separating the impact of exogenous nucleosides on RNA transcription and DNA replication.

Below I list what I believe are strengths and weaknesses in the manuscript:

Throughout the manuscript nucleoside and nucleotide are confused.

We thank the Reviewer for their feedback and thoughtful suggestions to improve the manuscript and for the time they took to provide comments. We have made every attempt to address the concerns, including performing LCMS-based absolute quantification of intracellular nucleotides and repeating experiments in multiple different cell lines – details are provided below. We have also worked to be more precise in our use of terminology in the revised manuscript.

Figure 1 - the experiment should be presented precisely, without assumptions. The data are interesting. Addition of nucleoside to the media impacts proliferation. You have no data

15as to whether that nucleoside impacts the concentration of nucleotides in the cell in Figure 1.

The Reviewer is correct that the data presented in Figure 1 do not address how addition of nucleotide precursors to the media impacts intracellular nucleotide levels. Indeed, this question is addressed by the data presented in Figure 2. We appreciate that this is a point of clarity and have updated the text in the revised manuscript accordingly.

Figure 2 - no standard curves are provided for the LC-MS/MS (?) used to quantitate intracellular nucleotide concentrations in cells. This is arguably the most important component of the manuscript and I find it wanting. What is the impact of nucleosides in the media on intracellular nucleotide concentrations in cells (and then does this impact transcription or replication). The authors state themselves that they cannot distinguish dGTP from ATP. No documentation about resolution and quantitation of nucleotides is provided. It is generally accepted that this is challenging. Details must be provided.

The data presented in Figure 2 of the original manuscript assessing nucleotide levels were indeed collected using LCMS, but only relative levels were measured. Thus, because the values presented were normalized peak areas, not absolute concentrations, the experiment did not include standard curves. Metabolites were identified based on an in-house database of masses and retention times measured on the same instruments, as stated in the Experimental Procedures section. They were also measured in a signal range we had previously determined to be in the linear range of detection for those species. We agree with the Reviewer that absolute quantification of nucleotides provides a deeper characterization of nucleotide imbalances caused by nucleotide precursor salvage. Therefore, for the revised manuscript we performed absolute quantification of intracellular nucleotides and present the measured nucleotide concentrations in Extended Data Fig. 2d-g in the revised manuscript (and the fold change in each nucleotide species is included as Fig. 2-g).

We also agree with the Reviewer that providing methods details is important. We apologize for any omissions and have updated the Experimental Procedures section of the paper to include technical details for all LCMS-based experiments, including the new absolute quantification experiments. We also provide raw data showing our standard curves for different nucleotides for the Reviewer to demonstrate that our detection of these metabolites is in the linear range (Reviewer Fig. 1).

16

Reviewer Figure 1. Standard curves for LCMS-based nucleotide quantification. Titrations of the indicated ¹³C/¹⁵N-labeled ribonucleotides (NTPs) (top) or dNTPs (bottom) were analyzed by LCMS. NTP standards were pooled together and dNTP standard were pooled together: because the intracellular levels of NTPs and dNTPs differ by an order of magnitude, NTPs were titrated up to 4 mM and dNTPs were titrated up to 400 μM. The correlation of standard concentration versus peak area measured by LCMS shows our detection of nucleotides in cells is in the linear range of detection for this assay. Additional methods details are provided in the Experimental Procedures section of the revised manuscript.

Figure 3 - it is reasonable to hypothesize that 200 uM G may impact the transport of EdU and subsequent intracellular concentration of EdU in the cell. This is ignored. There is a similar concern in Figure 6 where an ATRi is used (a competitive ATP inhibitor - an ATP analogue).

The Reviewer makes a good point; this is one reason why we assessed cell cycle state in multiple different ways that are not dependent on EdU transport during treatment with excess G. For example, we also examined DNA content to assess cell cycle state, since cells in S phase have intermediate DNA content between 2N and 4N. Additionally, we reasoned that a way to circumvent any potential transport issues is by performing an EdU pulse and then subsequently treating cells with G after washing out EdU, as presented in Fig. 3d,e and Extended Data Fig. 3f of the revised manuscript. These data also show that S phase progression is impaired by G supplementation. In addition, the fluorescent cell cycle reporter system confirms that S phase is extended in cells treated with G (Fig. 3f-i).

We also agree that it is ideal to confirm findings that rely on inhibitors with multiple drugs. To address this, we now include experiments with an additional ATR inhibitor that had the same effects as the ATR inhibitor used in the original manuscript (see Extended Data Fig. 6f in the revised manuscript).

Figure 4 is excellent. Nucleosides in the media do not appear to impact translation and growth. It would be helpful to look at transcription.

The Reviewer raises an interesting point about testing whether transcription is affected during nucleotide imbalance. We therefore examined cellular RNA levels following treatment with excess G as a way to test whether imbalance impairs global transcription; we found no major changes in RNA concentrations in cells (Reviewer Fig. 2). We are happy to include these data in the revised manuscript if the Reviewer or Editor feel strongly. Along with the observation that protein synthesis is not affected by nucleotide imbalance, this suggests that synthesis of cellular RNA is not impacted in a major way. This is an interesting point, as it indicates that ribonucleotide levels are still sufficient to sustain RNA synthesis even when they are imbalanced, and we discuss this point in the Discussion section of the revised manuscript.

Reviewer Figure 2. RNA concentrations in guanine-treated cells relative to untreated cells. A549 cells were cultured in standard culture conditions (Untreated) or in media containing 200 μ M guanine for 24 hours. Cells were then harvested and RNA levels were measured using a Qubit™ RNA High Sensitivity assay. RNA levels were then normalized to cell number and volume.

Figure 5 is excellent. Nucleosides in the media do impact DNA replication insofar as they induce DNA damage signaling that is generally (canonical signaling always) initiated at replication forks.

Figure 7 is excellent.

In summary, there are clear strengths in the paper and the ideas presented are provocative and interesting. However, the strengths are outweighed by the technical weaknesses, the absence of mechanism, and the use of a single cancer cell line. Finally, the manuscript is

18not lucid - it is actually quite hard to follow.

We thank the Reviewer for pointing out some strengths and apologize for any lack of clarity in our presentation. We have worked to correct this in the revised manuscript. We repeated key experiments in 3-5 cell lines and have added additional experiments in more cell lines with consistent findings in the revised manuscript. We also respectfully disagree that our findings lack mechanism. We believe the mechanism for how nucleotide imbalance uncouples proliferation from growth is as follows: imbalance is not sensed by metabolic regulatory pathways during G1 phase, which allows continued cell growth and allows cells to enter S phase. Then, imbalanced dNTPs induce replication stress to impair progression through S phase. We hope the Reviewer finds these points are more clearly articulated in the revised manuscript.

Reviewer #3:

Remarks to the Author:

In this manuscript, the authors show that several precursors of nucleotides can inhibit cell proliferation in a manner dependent on the salvage pathway. This observation is consistent with the known effects of thymidine. Furthermore, the authors show that guanine increases the synthesis of GTP through the salvage pathway but inhibits the de novo synthesis of both GTP and ATP, leading to an increase in GTP but a reduction in ATP and an imbalance between GTP and ATP. Adding adenine to cells restored ATP levels and reversed the GTP/ATP imbalance in guanine treated cells, and adenine also overcame the inhibition of cell proliferation by guanine.

The authors then carefully characterized how guanine affects cell proliferation during the cell cycle. They show that guanine clearly interferes with DNA replication in S phase and causes a delay in S/G2. Unexpectedly, however, guanine does not inhibit mTOR and protein synthesis, and cell growth is uncoupled from cell proliferation in cells with nucleotide imbalance. These results suggest that cells do not rely on the mTOR pathway to sense nucleotide imbalance.

Finally, the authors show that guanine treatment activates the ATR/ATM pathway in S phase, and that inhibition of ATR leads to irreversible DNA damage in cells with imbalanced nucleotides. They also provide evidence that ATR is activated during the unperturbed cell cycle, and that ATR promotes dNTP synthesis during S phase. Based on these results, the authors suggest that ATR plays a key role in sensing nucleotide imbalance and prevents the uncoupling of cell growth and cell proliferation.

This manuscript contains many interesting observations, and the model proposed is quite

19attractive. However, 'nucleotide imbalance' is not defined in a specific and quantitative way. Whether nucleotide imbalance or reduction in dNTP is the cause of replication problems and ATR activation is not clearly addressed. The known effects of thymidine and known functions of ATR in DNA replication also reduce the novelty of the model. Additional experiments are needed to strengthen this study and make it suitable for NCB.

We thank the Reviewer for their thoughtful evaluation of the data and for their recognition of the significance of the findings. We also thank them for their time considering our work. We appreciate that better defining nucleotide imbalance is important and have worked to do this in the revised manuscript, as well as add new data throughout to strengthen our conclusions.

Specific comments.

1. How 'nucleotide imbalance' is defined is not clear. As shown in Fig. 2d, many changes of nucleotides are observed after cells are treated with nucleotide precursors. What exactly is a 'nucleotide imbalance'? In the case of guanine, it is clear that the GTP/ATP balance is altered (Fig. 2a). However, in the case of thymidine, neither GTP/ATP nor UTP/CTP ratios change much, but cell proliferation is clearly inhibited (Fig. 2f). It is unclear to me whether 'nucleotide imbalance' refers to the imbalance between specific nucleotides or the overall imbalance among multiple nucleotides. Having a clear and quantitative definition of nucleotide imbalance is important for the model.

The Reviewer raises an excellent point, and we agree that more precisely quantifying how different disruptions to nucleotide levels impact proliferation lends deeper insight. To address this, we performed LCMS-based absolute quantification of intracellular nucleotide levels in cells treated with different nucleotide precursors to induce distinct imbalances in nucleotide levels. We present the absolute concentrations of intracellular nucleotides in Extended Data Fig. 2e-g of the revised manuscript. Because the baseline levels of each different intracellular nucleotide species vary over a wide range, we wanted to clearly understand the magnitude of change in levels of each species upon nucleotide precursor salvage. Therefore, we also used the absolute quantification data to calculate the fold-change in levels of each intracellular nucleotide species, and these data are presented as Fig. 2e-g of the revised manuscript.

To understand what degree of imbalance is needed to impair proliferation, we compared nucleotide levels in cells treated with concentrations of precursors that did or did not inhibit proliferation. In addition, we took advantage of the fact that different cell types have differential sensitivity to each precursor by comparing two cell lines with differential sensitivity to G and A nucleotides. As such, A549 cells are sensitive to 200 μ M G, while U2OS cells are not. Thus, we first tested how supplementation with 200 μ M G impacts intracellular nucleotide levels in these two

20cell lines. We then measured intracellular nucleotides in U2OS cells treated with 400 μM G, a dose that does inhibit proliferation in those cells. We compared the changes in nucleotide levels observed in each of these conditions. We find that at G concentrations that impact proliferation of each cell line, levels of GTP rise to 4-fold higher than untreated, while levels of ATP, UTP, and CTP fall to below 0.5-fold those observed in untreated cells. Additionally, levels of dTTP and dCTP fall to below 0.2-fold those of untreated cells. Conversely, U2OS cells are sensitive to 1.5 mM A, while A549 cells are not. We found that in each cell line, at concentrations of A that inhibit proliferation, ATP levels rise about 1.5-fold over normal, while UTP and CTP levels fall below .15-fold of normal. Additionally, dATP levels rise at least 1.5-fold over normal, and dTTP and dCTP levels fall below 0.4-fold of normal.

We expanded these analyses to perform a detailed characterization of cellular nucleotide levels in A549 cells treated with excess amounts of both purine and pyrimidine nucleotide precursors. The results from addition of purine nucleotides are described above. Addition of excess C caused CTP levels to rise about 15-fold, while causing GTP levels to drop to 0.5-fold of normal and UTP and ATP levels to drop to 0.25-fold of normal. Interestingly, dCTP levels were unchanged in these conditions, while dTTP levels were increased almost 3-fold, and dATP levels fell to 0.2-fold of normal. Treatment with excess T did not change ribonucleotide (NTP) balance, consistent with this nucleotide being exclusive to the dNTP pool. T treatment increased dTTP levels over 30-fold, increased dATP levels over 4-fold, and decreased dCTP levels below 0.1-fold.

We further evaluated which changes in nucleotide balance are important for proliferation by measuring nucleotide levels in cells where G- or T-induced proliferation arrest was rescued by providing A or C, respectively. We found that in these conditions, the imbalances in both NTPs and dNTPs were ameliorated.

Based on absolute quantification of nucleotide levels, we define nucleotide imbalance as an increase in one or more nucleotide species above normal levels along with a decrease in one or more other nucleotide species below normal levels that results in decreased proliferation without decreased cell growth. This is distinct from depletion of purines, pyrimidines or all NTP or dNTP species. We make this clear in the revised manuscript and incorporate the above-described data in support of this in Fig. 2e-g and Extended Data Fig. 2e-g. We have also edited the text in the revised manuscript to more precisely describe the changes in nucleotide balance that impact proliferation, and how these differ for each type of imbalance. Together, these data highlight that any imbalance in nucleotide levels that is significant enough can cause replication stress to impair proliferation without stopping cell growth. This emphasizes the important point that this phenotype is not dependent on depletion of any one particular nucleotide species.

2. In Fig. 2c and extended Fig. 2, A inhibited the salvage pathway for G, but G did not inhibit the salvage pathway for A. How can this difference between A and G be explained?

This is an intriguing observation that we also noticed. We think it is likely that salvage of adenine is more efficient than salvage of guanine based on the reported K_m values for their respective enzymes, APRT and HGPRT. Both enzymes use the substrate phosphoribosyl pyrophosphate substrate (PRPP) to produce the corresponding nucleotide, but APRT has a lower K_m for PRPP^{1,2}. Thus, although there is a small amount of G salvaged when A is also present, it is possible that APRT out-competes HGPRT for the PRPP substrate, and more A is salvaged than G.

3. Fig. 2a and 2e suggest that A reverses the effects of G on cell proliferation. Can one expect that G also reverses the effects of A on cell proliferation? Can the authors test whether G reverses the nucleotide changes caused by A?

Interestingly, we found that addition of G to cells in which proliferation is inhibited by A could not robustly rescue proliferation (Reviewer Fig. 3). Cells are sensitive to A at much higher concentrations than they are to G (Fig. 1b,f), consistent with the larger intracellular pools of A nucleotides. Thus, large excesses of A nucleotides may lead to imbalance that cannot be ameliorated even when G nucleotides are also salvaged. Indeed, we tested this condition in the LCMS-based quantification of nucleotide levels experiments and observed that adding G to cells treated with excess A did not restore nucleotide balance (see Extended Data Fig. 2d,e in the revised manuscript). Another possibility is that the feedback mechanisms that regulate ribonucleotide reductase are overwhelmed by excess nucleotides when large amounts of A are present, and normal dNTP balance cannot be reestablished.

Reviewer Figure 3. Addition of G to A-treated cells. A549 cells were supplemented with 2.5 mM adenine (A) with or without the addition of 25 μM guanine (G) as indicated.

4. In Fig. 2f, the rescue of T treated cells by C is not explained. Does C change any nucleotides affected by T? What is the key nucleotide imbalance that inhibits the proliferation of T treated cells? Thymidine is known to reduce dCTP. Is rescuing effect of C simply attributed to an increase of dCTP?

This is another interesting question. We quantified intracellular nucleotide levels in cells treated with excess T with or without rescue by addition of C (see Fig. 2d,e and Extended Data Fig. 2d,e). Cells treated with T alone had aberrantly increased dATP levels (as well as increased dTTP levels) in addition to decreased dCTP levels. Thus, T-treatment results in an imbalance of increased dATP and dTTP with decreased dCTP. Addition of C to T-treated cells did increase dCTP levels, but also restored balance of the other nucleotide species: addition of C ameliorated the excess dATP and also decreased excess dTTP levels by roughly two-fold. These data further support that cell growth and proliferation are decoupled by nucleotide imbalances involving the increase of at least one nucleotide species and the decrease of at least one nucleotide species. This is distinct from simple depletion of dNTPs. It is interesting to consider how the complex allosteric regulation of ribonucleotide reductase may contribute to dNTP imbalance under treatment with excess T (or other nucleotide precursors). Indeed, regulation of ribonucleotide reductase specificity for different nucleotides could help balance dNTP production during unperturbed proliferation but could exacerbate imbalances when nucleotide levels become imbalanced.

5. In Fig. 2g, it is clearly that guanine caused severe reductions in dTTP and dCTP. These changes are also observed in S phase after guanine treatment (Fig. 3k). Are these changes in dTTP and dCTP dependent on the salvage of G? Can these effects of G be reversed by A? If guanine reduces dTTP and dCTP, it becomes difficult to tell whether the S phase problems caused by guanine are attributed to nucleotide imbalance or the reduction in dNTPs. This is a conceptually important question to the model. If dNTP reduction is the cause of replication inhibition, the effects of guanine would become quite similar to those of thymidine, which is known to inhibit replication by reducing dCTP.

This is an important point, although we also find that thymidine increases dATP in addition to reducing dCTP (see Fig. 2e and Extended Data Fig. 2e in the revised manuscript), leading to an imbalance in dCTP relative to dATP and dTTP. To first address the Reviewer's experimental question, we examined NTP and dNTP levels in cells treated with excess G with or without rescue by addition of A. We found that providing G-treated cells with A did increase dTTP and dCTP levels (Fig. 2e and Extended Data Fig. 2e). Unfortunately, because dGTP has the same molecular

23weight and similar chromatographic properties to ATP (which is much more abundant), we could not reliably distinguish dGTP by LCMS. Nevertheless, it is likely that salvage of G increases dGTP levels due to increased levels of the ribonucleotide reductase substrate GDP. In this case, G salvage likely causes an imbalance of increased dGTP with decreased dCTP and dTTP (in addition to increased GTP with decreased ATP).

Moreover, other nucleotide precursors induce dNTP imbalances that do not solely deplete dTTP and dCTP (Fig. 2e and Extended Data Fig. 2e). This indicates that different types of nucleotide imbalances may ultimately become a problem when they lead to dNTP imbalance during S phase. While in each case relative depletion of one dNTP in relation to increased levels of another dNTP may cause inhibition of replication fork progression, it is important to consider that this is caused by salvage of NTP-precursors leading to imbalanced NTPs. This argues that cells do not have a robust mechanism to sense the relative levels of NTPs or dNTPs upon S phase entry and that this can lead to impaired S phase progression. Indeed, an important difference between effects of thymidine and effects of guanine is that guanine is salvaged to produce NTPs, while thymidine directly produces a dNTP. Thus, guanine salvage causes NTP imbalance upstream of any impact on dNTPs. We believe an important part of our findings is that NTP imbalance surprisingly does not impact biosynthesis and cell growth and is not detected by canonical growth regulatory pathways.

6. In Fig. 3j and 3k, one could argue that dTTP and dCTP are reduced because G treated cells did not progress through S phase efficiently. This appears to be a “chicken and egg” problem.

We agree that the relationship between dNTP levels and S phase progression can be complex because of the bidirectional connections. Nevertheless, because we find that nucleotides are imbalanced when cells enter S phase, it is likely that dNTP imbalance is causing S phase arrest, and not the other way around.

7. In Fig. 5d, it is surprising that LTX and BRQ did not activate ATR efficiently. LTX and BRQ can reduce DNA synthesis as efficiently as guanine (extend Fig. 3c). If DNA synthesis is severely compromised, why isn't ATR activated?

The Reviewer raises an interesting point. There is some activation of ATR present in LTX and BRQ-treated cells, consistent with a minor degree of replication stress. However, cells also respond to purine depletion via the mTORC1 pathway, which is consistent with prevention of S phase entry and therefore avoidance of replication stress. We suspect this may explain why these drugs do not activate ATR to the same extent as G treatment.

8. The data in Fig. 6 and 7 are consistent with the model in which ATR is important in cells with imbalanced nucleotides. However, dNTP levels are also changed in these cells. It is unclear whether nucleotide imbalance or reduction in dNTP is the cause of ATR dependency.

We agree that disruption to either one or all dNTP levels can cause cells to become dependent on ATR. While we agree that a relative decrease in levels of one or more nucleotides is an important part of the nucleotide imbalance phenotype, we found that for each imbalance, a relative decrease in at least one nucleotide species is accompanied by a relative increase in at least one other nucleotide species. This is distinct from depletion of one or more nucleotides; indeed, we find that treatments resulting in simple dNTP and NTP depletion do not cause ATR dependency (Extended Data Fig. 6g in the revised manuscript). We think that the ATR dependency observed during nucleotide imbalance is revealing because it shows that cells only become sensitive to the effects of imbalance once they enter S phase. Thus, even though NTP levels are disrupted, this does not inhibit RNA/protein synthesis or cell growth or perturb cell function during G1 phase. Instead, cells enter S phase despite imbalanced nucleotide levels and become dependent on ATR. Thus, in this case, imbalanced NTP levels are the upstream cause of ATR dependency in S phase. Importantly, this also argues that the primary sensor of imbalanced nucleotides is replication stress signaling, rather than canonical metabolic sensing pathways.

9. The novelty of Fig. 7 may be limited. Recent studies have shown that ATR plays an important role in unperturbed early S phase to limit replication origin firing and promote RRM2 accumulation (Buisson et al. Mol Cell 2015). The observations in Fig. 7 are quite consistent with the previous model and provide additional evidence on the changes of dNTPs in S phase.

We agree that elegant recent work has shown that ATR is activated during normal S phases. We are careful to reference that work in the revised manuscript and appreciate the Reviewer pointing out a paper that we previously neglected to cite. We completely agree that our findings are consistent with the importance of ATR activity in unperturbed S phases, and we show that ATR activity is specifically important for modulating the increase in dNTPs that occurs during normal S phase progression. These findings are complementary to recent publications, yet we also think that they provide additional support to a model in which the availability of balanced dNTPs is not regulated until cells enter S phase, consistent with our assertion that nucleotide imbalance is not sensed by upstream metabolic signaling and instead impacts S phase progression.

10. Thymidine is known to activate the salvage pathway and indirectly affect dCTP levels. Conceptually, this is quite similar to what is proposed in this study. Perhaps it is not

25surprising that nucleotide imbalance would affect dNTP levels and indirectly cause replication stress and inhibition of cell proliferation. The novel finding of this study is that nucleotide imbalance is not detected by the mTOR pathway, unlike the depletion of nucleotides. It would be important to better explain how even imbalanced nucleotides can activate mTOR and how mTOR promotes S phase entry in this context.

The Reviewer refers to the important point that the data show mTORC1 signaling does not sense imbalanced nucleotides. We confirmed that mTORC1 is inhibited by purine nucleotide depletion, consistent with prior observations. We took advantage of this to test whether providing A or G in amounts that either do or do not lead to nucleotide imbalance could differentially restore mTORC1 signaling in purine-deprived cells. Interestingly, we found that supplementing purine-deprived cells with levels of A or G that either do or do not cause nucleotide imbalance restored mTORC1 signaling (Extended Data Fig. 4l,m of the revised manuscript). These data suggest that the presence of adequate amounts of either A- or G- nucleotides, even if they are imbalanced, is sufficient to maintain mTORC1 activity. Thus, when cells are supplemented with excess G or A, high intracellular levels of either G- or A- nucleotides can maintain growth signaling even while the resulting nucleotide imbalance inhibits proliferation (Extended Data Fig. 4n in the revised manuscript). mTORC1 activity has been shown to promote S phase entry through its downstream effectors S6K1 and eIF4E, while mTORC1 inhibition causes cells to stall in G1 phase^{3,4}. Indeed, we found that supplementing purine-deprived cells with levels of G that induce imbalance but restore mTORC1 signaling enables S phase entry and subsequently causes S phase stalling (see Extended Fig. 4o in the revised manuscript). Conversely, pharmacologically inhibiting mTORC1 in cells treated with excess G prevented S phase entry (Extended Data Fig. 4o in the revised manuscript), suggesting that mTORC1 activity is important for allowing S phase entry during nucleotide imbalance. These points are discussed in the revised manuscript.

References

1. Free, M. L. *et al.* Expression of active human hypoxanthine-guanine phosphoribosyltransferase in Escherichia coli and characterisation of the recombinant enzyme. *Biochim. Biophys. Acta BBA - Gene Struct. Expr.* **1087**, 205–211 (1990).
2. Huyet, J. *et al.* Structural Insights into the Forward and Reverse Enzymatic Reactions in Human Adenine Phosphoribosyltransferase. *Cell Chem. Biol.* **25**, 666-676.e4 (2018).
3. Fingar, D. C. *et al.* mTOR Controls Cell Cycle Progression through Its Cell Growth Effectors S6K1 and 4E-BP1/Eukaryotic Translation Initiation Factor 4E. *Mol. Cell. Biol.* **24**, 200–216 (2004).

4. Cuyàs, E., Corominas-Faja, B., Joven, J. & Menendez, J. A. Cell Cycle Regulation by the Nutrient-Sensing Mammalian Target of Rapamycin (mTOR) Pathway. in *Cell Cycle Control* (eds. Noguchi, E. & Gadaleta, M. C.) vol. 1170 113–144 (Springer New York, 2014).

Decision Letter, first revision:

3rd May 2022

Dear Dr. Vander Heiden,

Thank you for submitting your revised manuscript "Nucleotide imbalance decouples cell growth from cell proliferation" (NCB-V45889A). It has now been seen by the original referees and their comments are below. Reviewers #1 and #3 agreed to assess the responses to Rev#2, who was not available to re-review. The reviewers both found that the paper has improved in revision, including in addressing Rev#2's points, and therefore we'll be happy in principle to publish it in *Nature Cell Biology*, pending minor revisions to satisfy the referees' final requests and to comply with our editorial and formatting guidelines.

Please note that the current version of your manuscript is in a PDF format, could you please email us a copy of the file in an editable format (Microsoft Word or LaTeX)? We unfortunately cannot proceed with PDFs at this stage.

After receiving the Word file, we will perform detailed checks on your paper and will send you a checklist detailing our editorial and formatting requirements in about a week. Please do not upload the final materials and make any revisions until you receive this additional information from us.

Thank you again for your interest in *Nature Cell Biology*. Please do not hesitate to contact me if you have any questions.

Sincerely,

Melina

Melina Casadio, PhD
Senior Editor, *Nature Cell Biology*
ORCID ID: <https://orcid.org/0000-0003-2389-2243>

Reviewer #1 (Remarks to the Author):

The authors have addressed my concerns.

27Reviewer #3 (Remarks to the Author):

The authors have done a good job in addressing my prior comments. In particular, the new LCMS data added in Fig. 2e-g and Extended Fig. 2e-g have significantly strengthened the model of this study. The clear definition of nucleotide imbalance will help readers better understand the concept and test the model in future studies. However, a few minor issues arose from these new data. I hope that the authors can clarify.

1. In Fig. 2d, T appears to be different from other precursors. It did not cause an imbalance of NTP by the new definition. However, T clearly reduces cell proliferation (Fig. 1c) and activates the ATR response (Fig. 5c). Are the effects of T independent of NTP imbalance?
2. In Fig. 2f, the G-induced NTP imbalance in A549 cells but not U2OS cells nicely explains why A549 cells are more sensitive to G than U2OS cells. However, the A-induced NTP imbalance is similar in A549 and U2OS cells. This won't explain why U2OS cells are more sensitive to A. Can the authors explain this?
3. In Fig. 4a, it seems that mTOR remains active after BRQ treatment. Would it suggest that pyrimidine depletion is not detected by mTOR? Furthermore, would it argue that the decoupling of DNA synthesis and cell growth is not unique to NTP imbalance? It may be an important point to clarify in the model.

10th May 2022

Dear Dr. Vander Heiden,

Thank you for your patience as we've prepared the guidelines for final submission of your Nature Cell Biology manuscript, "Nucleotide imbalance decouples cell growth from cell proliferation" (NCB-V45889A). Please carefully follow the step-by-step instructions provided in the attached file, and add a response in each row of the table to indicate the changes that you have made. Please also check and comment on any additional marked-up edits we have proposed within the text. Ensuring that each point is addressed will help to ensure that your revised manuscript can be swiftly handed over to our production team.

We would like to start working on your revised paper, with all of the requested files and forms, as soon as possible (preferably within one week). Please get in contact with us if you anticipate delays.

28In recognition of the time and expertise our reviewers provide to Nature Cell Biology's editorial process, we would like to formally acknowledge their contribution to the external peer review of your manuscript entitled "Nucleotide imbalance decouples cell growth from cell proliferation". For those reviewers who give their assent, we will be publishing their names alongside the published article.

Nature Cell Biology offers a Transparent Peer Review option for new original research manuscripts submitted after December 1st, 2019. As part of this initiative, we encourage our authors to support increased transparency into the peer review process by agreeing to have the reviewer comments, author rebuttal letters, and editorial decision letters published as a Supplementary item. When you submit your final files please clearly state in your cover letter whether or not you would like to participate in this initiative. Please note that failure to state your preference will result in delays in accepting your manuscript for publication.

Cover suggestions

As you prepare your final files we encourage you to consider whether you have any images or illustrations that may be appropriate for use on the cover of Nature Cell Biology.

Nature Cell Biology has now transitioned to a unified Rights Collection system which will allow our Author Services team to quickly and easily collect the rights and permissions required to publish your work. Approximately 10 days after your paper is formally accepted, you will receive an email in providing you with a link to complete the grant of rights. If your paper is eligible for Open Access, our Author Services team will also be in touch regarding any additional information that may be required to arrange payment for your article.

Please note that *Nature Cell Biology* is a Transformative Journal (TJ). Authors may publish their research with us through the traditional subscription access route or make their paper immediately open access through payment of an article-processing charge (APC). Authors will not be required to make a final decision about access to their article until it has been accepted. Find out more about Transformative Journals

Please use the following link for uploading these materials:
[REDACTED]

Best regards,

Nyx Hills
Staff
Nature Cell Biology

On behalf of

Melina Casadio, PhD
Senior Editor, Nature Cell Biology
ORCID ID: <https://orcid.org/0000-0003-2389-2243>

Reviewer #1:
Remarks to the Author:
The authors have addressed my concerns.

Reviewer #3:

30Remarks to the Author:

The authors have done a good job in addressing my prior comments. In particular, the new LCMS data added in Fig. 2e-g and Extended Fig. 2e-g have significantly strengthened the model of this study. The clear definition of nucleotide imbalance will help readers better understand the concept and test the model in future studies. However, a few minor issues arose from these new data. I hope that the authors can clarify.

1. In Fig. 2d, T appears to be different from other precursors. It did not cause an imbalance of NTP by the new definition. However, T clearly reduces cell proliferation (Fig. 1c) and activates the ATR response (Fig. 5c). Are the effects of T independent of NTP imbalance?
2. In Fig. 2f, the G-induced NTP imbalance in A549 cells but not U2OS cells nicely explains why A549 cells are more sensitive to G than U2OS cells. However, the A-induced NTP imbalance is similar in A549 and U2OS cells. This won't explain why U2OS cells are more sensitive to A. Can the authors explain this?
3. In Fig. 4a, it seems that mTOR remains active after BRQ treatment. Would it suggest that pyrimidine depletion is not detected by mTOR? Furthermore, would it argue that the decoupling of DNA synthesis and cell growth is not unique to NTP imbalance? It may be an important point to clarify in the model.

Author Rebuttal, first revision:

Reviewer comments in ***bold italics*** and our response in plain text:

Reviewer 3:

The authors have done a good job in addressing my prior comments. In particular, the new LCMS data added in Fig. 2e-g and Extended Fig. 2e-g have significantly strengthened the model of this study. The clear definition of nucleotide imbalance will help readers better understand the concept and test the model in future studies. However, a few minor issues arose from these new data. I hope that the authors can clarify.

We thank the Reviewer for their time spent reading our revised manuscript. We agree that including these new data improved the manuscript.

1. ***In Fig. 2d, T appears to be different from other precursors. It did not cause an imbalance of NTP by the new definition. However, T clearly reduces cell proliferation (Fig. 1c) and activates the ATR response (Fig. 5c). Are the effects of T independent of NTP imbalance?***

The Reviewer raises a good point. Because thymidine is unique to the dNTP pool, excess thymidine salvage causes imbalance only among dNTP species, without impacting NTP balance. This is indeed slightly

31different from the consequences of other nucleotide precursors, which directly cause NTP imbalance upstream of changes to dNTP balance. Ultimately however, the dNTP imbalance resulting either from salvage of thymidine or from salvage of other NTP precursors impairs proliferation via replication stress and S phase stalling. Thus, while the same mechanism ultimately inhibits proliferation, we agree that it is important to distinguish the direct effect of thymidine on dNTPs, and we now clarify this point in the revised manuscript. Indeed, that DNA replication is impaired following NTP imbalance in addition to direct dNTP perturbations emphasizes the point that cells lack a mechanism to sense nucleotide balance prior to S phase entry.

- 2. In Fig. 2f, the G-induced NTP imbalance in A549 cells but not U2OS cells nicely explains why A549 cells are more sensitive to G than U2OS cells. However, the A- induced NTP imbalance is similar in A549 and U2OS cells. This won't explain why U2OS cells are more sensitive to A. Can the authors explain this?**

Upon supplementation with 1.5 mM adenine, levels of UTP, CTP, dTTP, and dCTP do not decrease to the same extent in A549 cells as they do in U2OS cells. This cell type- specific difference in nucleotide levels is not as striking at the difference upon 200 μ M guanine addition, but has a meaningful impact on proliferation. Thus, these changes to nucleotide balance in A549 cells may be better tolerated because they still allow sufficient dNTPs for replication, while the more extreme imbalance in U2OS cells may cross a threshold of imbalance that can no longer support efficient replication.

- 3. In Fig. 4a, it seems that mTOR remains active after BRQ treatment. Would it suggest that pyrimidine depletion is not detected by mTOR? Furthermore, would it argue that the decoupling of DNA synthesis and cell growth is not unique to NTP imbalance? It may be an important point to clarify in the model.**

This is an interesting point, and we agree that the lack of mTOR response to BRQ treatment does suggest that pyrimidine depletion is not sensed by mTOR in the same way as purine depletion. We also agree that interestingly, this suggests that decoupling of cell growth and proliferation may be implicated during other metabolic perturbations and could play a wider role in how cells recover from other metabolic stressors. We now discuss the lack of mTORC1 response to BRQ in t

Final Decision Letter:

Dear Dr Vander Heiden,

I am pleased to inform you that your manuscript, "Nucleotide imbalance decouples cell growth from cell proliferation", has now been accepted for publication in Nature Cell Biology. Congratulations on this very interesting study!

Please note that *Nature Cell Biology* is a Transformative Journal (TJ). Authors may publish their research with us through the traditional subscription access route or make their paper immediately open access through payment of an article-processing charge (APC). Authors will not be required to make a final decision about access to their article until it has been accepted. Find out more about

2Transformative Journals

If you have not already done so, we strongly recommend that you upload the step-by-step protocols used in this manuscript to the Protocol Exchange (www.nature.com/protocolexchange), an open online resource established by Nature Protocols that allows researchers to share their detailed experimental know-how. All uploaded protocols are made freely available, assigned DOIs for ease of citation and are fully searchable through nature.com. Protocols and Nature Portfolio journal papers in which they are used can be linked to one another, and this link is clearly and prominently visible in the online versions of both papers. Authors who performed the specific experiments can act as primary authors for the Protocol as they will be best placed to share the methodology details, but the Corresponding Author of the present research paper should be included as one of the authors. By uploading your Protocols to Protocol Exchange, you are enabling researchers to more readily reproduce or adapt the methodology you use, as well as increasing the visibility of your protocols and papers. You can also establish a dedicated page to collect your lab Protocols. Further information can be found at www.nature.com/protocolexchange/about

With kind regards,

Melina

Melina Casadio, PhD
Senior Editor, Nature Cell Biology
ORCID ID: <https://orcid.org/0000-0003-2389-2243>

** Visit the Springer Nature Editorial and Publishing website at www.springernature.com/editorial-and-publishing-jobs for more information about our career opportunities. If you have any questions please click here.**